# Habitat 2.0:
# Training Home Assistants to Rearrange their Habitat

**Andrew Szot[2], \*Alex Clegg[1], Eric Undersander[1], Erik Wijmans[1,2], Yili Zhao[1], John Turner[1],
Noah Maestre[1], Mustafa Mukadam[1], Devendra Chaplot[1], Oleksandr Maksymets[1],
Aaron Gokaslan[1], Vladimir Vondrus, Sameer Dharur[2], Franziska Meier[1], Wojciech Galuba[1],
Angel Chang[4], Zsolt Kira[2], Vladlen Koltun[3], Jitendra Malik[1,5], Manolis Savva[4], Dhruv Batra[1,2]**
[1]Facebook AI Research, [2]Georgia Tech, [3]Intel Research, [4]Simon Fraser University [5]UC Berkeley

## Abstract

We introduce Habitat 2.0 (H2.0), a simulation platform for training virtual robots in *interactive* 3D environments and complex physics-enabled scenarios. We make comprehensive contributions to all levels of the embodied AI stack – data, simulation, and benchmark tasks. Specifically, we present: (i) ReplicaCAD: an artist-authored, annotated, reconfigurable 3D dataset of apartments (matching real spaces) with articulated objects (*e.g.* cabinets and drawers that can open/close); (ii) H2.0: a high-performance physics-enabled 3D simulator with **speeds exceeding 25,000 simulation steps per second ($850\times$ real-time)** on an 8-GPU node, representing $100\times$ speed-ups over prior work; and, (iii) Home Assistant Benchmark (HAB): a suite of common tasks for assistive robots (tidy the house, stock groceries, set the table) that test a range of mobile manipulation capabilities. These large-scale engineering contributions allow us to systematically compare deep reinforcement learning (RL) at scale and classical sense-plan-act (SPA) pipelines in long-horizon structured tasks, with an emphasis on generalization to new objects, receptacles, and layouts. We find that (1) flat RL policies struggle on HAB compared to hierarchical ones; (2) a hierarchy with independent skills suffers from 'hand-off problems', and (3) SPA pipelines are more brittle than RL policies.

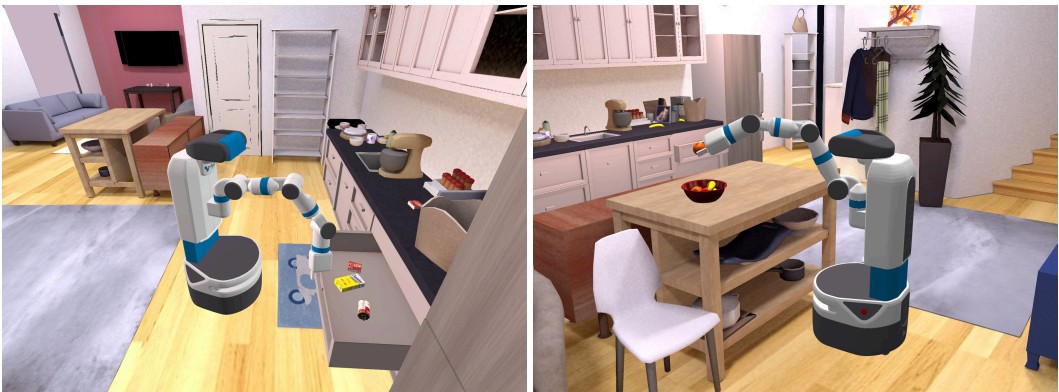

Figure 1: A mobile manipulator (Fetch robot) simulated in Habitat 2.0 performing rearrangement tasks in a ReplicaCAD apartment – (left) opening a drawer before picking up an item from it, and (right) placing an object into the bowl after navigating to the table. Best viewed in motion at https://aihabitat.org/docs/habitat2.

## 1   Introduction

Consider a home assistant robot illustrated in Fig. 1 – a mobile manipulator (Fetch [1]) performing tasks like stocking groceries into the fridge, clearing the table and putting dishes into the dishwasher,

---

*Correspondence to: `aszot3@gatech.edu`

35th Conference on Neural Information Processing Systems (NeurIPS 2021).

fetching objects on command and putting them back, *etc*. Developing such embodied intelligent systems is a goal of deep scientific and societal value. So how should we accomplish this goal?

Training and testing such robots in hardware directly is slow, expensive, and difficult to reproduce. We aim to advance the entire 'research stack' for developing such embodied agents in simulation – (1) data: curating house-scale interactive 3D assets (*e.g.* kitchens with cabinets, drawers, fridges that can open/close) that support studying generalization to unseen objects, receptacles, and home layouts, (2) simulation: developing the next generation of high-performance photo-realistic 3D simulators that support rich interactive environments, (3) tasks: setting up challenging representative benchmarks to enable reproducible comparisons and systematic tracking of progress over the years. To support this long-term research agenda, we present:

• **ReplicaCAD**: an artist-authored fully-interactive recreation of 'FRL-apartment' spaces from the Replica dataset [2] consisting of 111 unique layouts of a single apartment background with 92 authored objects including dynamic parameters, semantic class and surface annotations, and efficient collision proxies, representing 900+ person-hours of professional 3D artist effort. ReplicaCAD (illustrated in figures and videos) was created with the consent of and compensation to artists, and will be shared under a Creative Commons license for non-commercial use with attribution (CC-BY-NC).

• **Habitat 2.0 (H2.0)**: a high-performance physics-enabled 3D simulator, representing approximately 2 years of development effort and the next generation of the Habitat project [3] (Habitat 1. 0). H2.0 supports piecewise-rigid objects (*e.g.* door, cabinets, and drawers that can rotate about an axis or slide), articulated robots (*e.g.* mobile manipulators like Fetch [1], fixed-base arms like Franka [4], quadrupeds like AlienGo [5]), and rigid-body mechanics (kinematics and dynamics). The design philosophy of H2.0 is to prioritize performance (or speed) over the breadth of simulation capabilities. H2.0 by design and choice does not support non-rigid dynamics (deformables, fluids, films, cloths, ropes), physical state transformations (cutting, drilling, welding, melting), audio or tactile sensing – many of which are capabilities provided by other simulators [6–8]. The benefit of this focus is that we were able to design and optimize H2.0 to be *exceedingly* fast – simulating a Fetch robot interacting in ReplicaCAD scenes at 1200 steps per second (SPS), where each 'step' involves rendering 1 RGBD observation (128×128 pixels) and simulating rigid-body dynamics for $1/30$ sec. Thus, 30 SPS would be considered 'real time' and 1200 SPS is 40× real-time. H2.0 also scales well – achieving 8,200 SPS (273× real-time) multi-process on a single GPU and over 25,000 SPS (850× real-time) on a single node with 8 GPUs. For reference, existing simulators typically achieve 10-400 SPS (see Tab. 1). These 100× simulation-speedups correspond to cutting experimentation time from 6 months to under 2 days, unlocking experiments that were hitherto infeasible, allowing us to answer questions that were hitherto unanswerable. As we will show, they also directly translate to training-time speed-up and accuracy improvements from training agents (for object rearrangement tasks) on more experience.

• **Home Assistant Benchmark (HAB):** a suite of common tasks for assistive robots (`TidyHouse`, `PrepareGroceries`, `SetTable`) that are specific instantiations of the generalized rearrangement problem [9]. Specifically, a mobile manipulator (Fetch) is asked to rearrange a list of objects from initial to desired positions – picking/placing objects from receptacles (counter, sink, sofa, table), opening/closing containers (drawers, fridges) as necessary. We use the GeometricGoal specification prescribed by Batra *et al*. [9] – *i.e.*, initial and desired 3D (center-of-mass) position of each target object $i$ to be rearranged $\left(s_i^0, s_i^*\right)_{i=1}^N$. The choice of GeometricGoal is deliberate – we aim to create the PointNav [10] equivalent for mobile manipulators. As witnessed in the navigation literature, such a task becomes the testbed for exploring ideas [11–19] and a starting point for more semantic tasks [20–22]. The robot operates entirely from onboard sensing – head- and arm-mounted RGB-D cameras, proprioceptive joint-position sensors (for the arm), and egomotion sensors (for the mobile base) – and may not access any privileged state information (no prebuilt maps, no 3D models of rooms or objects, no physically-implausible sensors providing knowledge of mass, friction, articulation of containers, *etc*.). Notice that an object's center-of-mass provides no information about its size or orientation. The target object may be located inside a container (drawer, fridge), on top of supporting surfaces (shelf, table, sofa) of varying heights and sizes, and surrounded by clutter; all of which must be sensed and maneuvered. Receptacles like drawers and fridges start closed, meaning that the agent must open and close articulated objects to succeed. An episode is considered successful if all target objects are placed within 15cm of their desired positions (without considering orientation). The robot uses continuous end-effector control for the arm and velocity control for the base. We deliberately focus on gross motor control (the base and arm) and not fine motor control (the gripper), following

| | Rendering | | Physics | | Scene | Speed |
|---|---|---|---|---|---|---|
| | Library | Supports | Library | Supports | Complexity | (steps/sec) |
| Habitat [3] | Magnum | 3D scans | none | continuous navigation (navmesh) | building-scale | 3,000 |
| AI2-THOR [6] | Unity | Unity | Unity | rigid dynamics, animated interactions | room-scale | 30 - 60 |
| ManipulaTHOR [33] | Unity | Unity | Unity | AI2-THOR + manipulation | room-scale | 30 - 40 |
| ThreeDWorld [7] | Unity | Unity | Unity (PhysX) + FLEX | rigid + particle dynamics | room/house-scale | 5 - 168 |
| SAPIEN [34] | OpenGL/OptiX | configurable | PhysX | rigid/articulated dynamics | object-level | 200 - 400$^\dagger$ |
| RLBench [35] | CoppeliaSim (OpenGL) | Gouraud shading | CoppeliaSim (Bullet/ODE) | rigid/articulated dynamics | table-top | 1 - 60$^\dagger$ |
| iGibson [36] | PyRender | PBR shading | PyBullet | rigid/articulated dynamics | house-scale | 100 |
| Habitat 2.0 (H2.0) | Magnum | 3D scans + PBR shading | Bullet | rigid/articulated dynamics + navmesh | house-scale | 1,200 |

Table 1: High-level comparison of different simulators. Note: Speeds were taken directly from respective publications or obtained via direct personal correspondence with the authors when not publicly available (indicated by $^\dagger$). Benchmarking was conducted by different teams on different hardware with different underlying 3D assets simulating different capabilities. Thus, these should be considered qualitative comparisons representing what a user expects to experience on a single instance of the simulator (no parallelization).

the 'abstracted grasping' recommendations from [9]. Specifically, once the end-effector reaches 15cm (or closer) to an object, a discrete grasp action becomes available that, if executed, snaps the object into its parallel-jaw gripper [2]. We conduct a systematic study of two distinct techniques – monolithic 'sensors-to-actions' policies trained with reinforcement learning (RL) at scale, and classical sense-plan-act pipelines (SPA) [26] – with a particular emphasis on systematic generalization to new objects, receptacles, apartment layouts (not just robot starting pose). Our findings include:

1. **Flat vs hierarchical:** Monolithic RL policies successfully learn diverse *individual* skills (pick/place, navigate, open/close drawer). However, crafting a combined reward function and learning scheme that elicits chaining of such skills for the long-horizon HAB tasks remained out of our reach. We saw significantly stronger results with a hierarchical approach that assumes knowledge of a perfect task planner (via STRIPS [27]) to break it down into a sequence of skills.
2. **Hierarchy cuts both ways:** However, a hierarchy with independent skills suffers from 'hand-off problems' where a succeeding skill isn't set up for success by the preceding one – *e.g.*, navigating to a bad location for subsequent manipulation, only partially opening a drawer to grab an object inside, or knocking an object out of reach that is later needed.
3. **Brittleness of SensePlanAct:** For simple skills, SPA performs just as well as monolithic RL. However, it is significantly more brittle since it needs to map all obstacles in the workspace for planning. More complex settings involving clutter, challenging receptacles, and imperfect navigation can poorly frame the target object and obstacles in the robot's camera, leading to incorrect plans.

We hope our work will serve as a benchmark for many years to come. H2.0 is free, open-sourced under the MIT license, and under active development. [3] We believe it will reduce the community's reliance on commercial lock-ins [28, 29] and non-photorealistic simulation engines [30–32].

## 2  Related Work

**What *is* a simulator?** Abstractly speaking, a simulator has two components: (1) a *physics engine* that evolves the world state $s$ over time $s_t \to s_{t+1}$, and (2) a *renderer* that generates sensor observations $o$ from states: $s_t \to o_t$. The boundary between the two is often blurred as a matter of convenience. Many physics engines implement minimal renderers to visualize results, and some rendering engines include integrations with a physics engine. PyBullet [37], MuJoCo [28], DART [38], ODE [39], PhysX/FleX [40, 41], and Chrono [42] are primarily physics engines with some level of rendering, while Magnum [43], ORRB [44], and PyRender [45] are primarily renderers. Game engines like Unity [46] and Unreal [47] provide tightly coupled integration of physics and rendering. Some simulators [3, 48, 49] involve largely static environments – the agent can move but not change the state of the environment (*e.g.* open cabinets). Thus, they are heavily invested in rendering with fairly lightweight physics (*e.g.* collision checking with the agent approximated as a cylinder).

**How are interactive simulators built today?** Either by relying on game engines [6, 50, 51] or via a 'homebrew' integration of existing rendering and physics libraries [7, 34, 36, 52]. Both options have problems. Game engines tend to be optimized for human needs (high image-resolution, ∼60 FPS, persistent display) not for AI's needs [53] (10k+ FPS, low-res, 'headless' deployment on a

---

[2]To be clear, H2.0 fully supports the rigid-body mechanics of grasping; the abstract grasping is a *task-level* simplification that can be trivially undone. Grasping, in-hand manipulation, and goal-directed releasing of a grasp are all challenging open research problems [23–25] that we believe must further mature in the fixed-based close-range setting before being integrated into a long-horizon home-scale rearrangement problem.

[3]Start using Habitat 2.0 with the tutorial at https://aihabitat.org/docs/habitat2

cluster). Reliance on them leads to limited control over the performance characteristics. On the other hand, they represent decades of knowledge and engineering effort whose value cannot be discounted. This is perhaps why 'homebrew' efforts involve a high-level (typically Python-based) integration of existing libraries. Unfortunately but understandably, this results in simulation speeds of 10-100s of SPS, which is *orders of magnitude* sub-optimal. H2.0 involved a deep low-level (C++) integration of rendering (via Magnum [43]) and physics (via Bullet [37]), enabling precise control of scheduling and task-aware optimizations, resulting in substantial performance improvements.

**Object rearrangement.** Task- and motion-planning [54] and mobile manipulation have a long history in AI and robotics, whose full survey is beyond the scope of this document. Batra et al. [9] provide a good summary of the historical background of rearrangement, a review of recent efforts, a general framework, and a set of recommendations that we adopt here. Broadly speaking, our work is distinguished from prior literature by a combination of the emphasis on visual perception, lack of access to state, systematic generalization, and the experimental setup of visually-complex and ecologically-realistic home-scale environments. We now situate w.r.t. a few recent efforts. [55] study replanning in the presence of partial observability but do not consider mobile manipulation. [52] tackle 'interactive navigation', where the robot can bump into and push objects during navigation, but does not have an arm. Some works [56–58] abstract away gross motor control entirely by using symbolic interaction capabilities (*e.g.* a 'pick up X' action) or a 'magic pointer' [9]. We use abstracted grasping but not abstract manipulation. [19] develop hierarchical methods for mobile manipulation, combining RL policies for goal-generation and motion-planning for executing them. We use the opposite combination of planning and learning – using task-planning to generate goals and RL for skills. [33] is perhaps the most similar to our work. Their task involves moving a single object from one location to another, excluding interactions with container objects (opening a drawer or fridge to place an object inside). We will see that rearrangement of multiple objects while handling containment is a much more challenging task. Interestingly, our experiments show evidence for the opposite conclusion reached therein – monolithic end-to-end trained RL methods are outperformed by a modular approach that is trained stage-wise to handle long-horizon rearrangement tasks.

## 3 Replica to ReplicaCAD: Creating Interactive Digital Twins of Real Spaces

We begin by describing our dataset that provides a rich set of indoor layouts for studying rearrangement tasks. Our starting point was Replica [2], a dataset of *highly* photo-realistic 3D reconstructions at room and building scale. Unfortunately, static 3D scans are unsuitable for studying rearrangement tasks because objects in a static scan cannot be moved or manipulated.

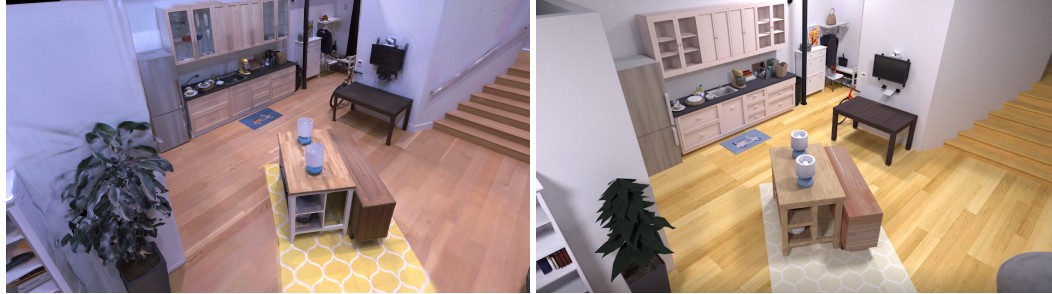

Figure 2: Left: The original Replica scene. Right: the artist recreated scene ReplicaCAD. All objects (furniture, mugs) including articulated ones (drawers, fridge) in ReplicaCAD are fully physically simulated and interactive.

**Asset Creation.** ReplicaCAD is an artist-created, fully-interactive recreation of 'FRL-apartment' spaces from the Replica dataset [2]. First, a team of 3D artists authored individual 3D models (geometry, textures, and material specifications) to faithfully recreate nearly all objects (furniture, kitchen utensils, books, *etc*.; 92 in total) in all 6 rooms from the FRL-apartment spaces as well as an accompanying static backdrop (floor and walls). Fig. 2 compares a layout of ReplicaCAD with the original Replica scan. Next, each object was prepared for rigid-body simulation by authoring physical parameters (mass, friction, restitution), collision proxy shapes, and semantic annotations. Several objects (*e.g.* refrigerator, kitchen counter) were made 'articulated' through sub-part segmentation (annotating fridge door, counter cabinet) and authoring of URDF files describing joint configurations (*e.g.* fridge door swings around a hinge) and dynamic properties (*e.g.* joint type and limits). For each

large furniture object (*e.g.* table), we annotated surface regions (*e.g.* table tops) and containment volumes (*e.g.* drawer space) to enable programmatic placement of small objects on top of or within.

**Human Layout Generation.** Next, a 3D artist authored an additional 5 semantically plausible 'macro variations' of the scenes – producing new scene layouts consisting only of larger furniture from the same 3D object assets. Each of these macro variations was further perturbed through 20 'micro variations' that re-positioned objects – *e.g.* swapping the locations of similarly sized tables or a sofa and two chairs. This resulted in a total of 105 scene layouts that exhibit major and minor semantically-meaningful variations in furniture placement and scene layout, enabling controlled testing of generalization. Illustrations of these variations can be found in Appendix A.

**Procedural Clutter Generation.** To maximize the value of the human-authored assets we also develop a pipeline that allows us to generate new clutter procedurally. Specifically, we dynamically populate the annotated supporting surfaces (*e.g.* table-top, shelves in a cabinet) and containment volumes (*e.g.* fridge interior, drawer spaces) with object instances from appropriate categories (*e.g.*, plates, food items). These inserted objects can come from ReplicaCAD or the YCB dataset [59]. We compute physically-stable insertions of clutter offline (*i.e.* letting an inserted bowl 'settle' on a shelf) and then load these stable arrangements into the scene dynamically at run-time.

ReplicaCAD is fully integrated with the H2.0 and a supporting configuration file structure enables simple import, instancing, and programmatic alternation of any of these interactive scenes. Overall, ReplicaCAD represents 900+ person-hours of professional 3D artist effort so far (with augmentations in progress). It was created with the consent of and compensation to artists, and will be shared under a Creative Commons license for non-commercial use with attribution (CC-BY-NC). Further ReplicaCAD details and statistics are in Appendix A.

## 4 Habitat 2.0 (H2.0): a Lazy Simulator

H2.0's design philosophy is that speed is more important than the breadth of capabilities. H2.0 achieves fast rigid-body simulation in large photo-realistic 3D scenes by being lazy and only simulating what is absolutely needed. We instantiate this principle via 3 key ideas – localized physics and rendering (Sec. 4.1), interleaved physics and rendering (Sec. 4.2), and simplify-and-reuse (Sec. B.1). We also describe motion planning integration in Appendix B.2.

### 4.1 Localized Physics and Rendering

Realistic indoor 3D scenes can span houses with multiple rooms (kitchen, living room), hundreds of objects (sofa, table, mug) and 'containers' (fridge, drawer, cabinet), and thousands of parts (fridge shelf, cabinet door). Simulating physics for every part at all times is slow and unnecessary. We leverage Bullet's built-in island sleep system to minimize simulation overhead for idle objects. In addition, we make several optimizations: (1) We employ a navigation mesh to move the robot base kinematically (which has been shown to transfer well to real the world [60]) rather than simulating wheel-ground contact. (2) For multi-body articulated furniture, we remove static parts (e.g. the walls and floor of a cabinet) from the Bullet multi-body and instead load these as separate static rigid objects. This improves the sleeping behavior of the entire simulation, for example, an idle object resting on the floor of the cabinet can sleep even while the cabinet door is moving. (3) We use the sleeping state of objects to optimize rendering by caching and re-using scene graph transformation matrices and frustum-culling results.

### 4.2 Interleaved rendering and physics

Most physics engines (*e.g.* Bullet) run on the CPU, while rendering (*e.g.* via Magnum) typically occurs on the GPU. After our initial optimizations, we found each to take nearly equal compute-time. This represents a *glaring* inefficiency – as illustrated in Fig. 3, at any given time either the CPU is sitting idle waiting for the GPU or vice-versa. Thus, interleaving them leads to significant gains. However, this is complicated by a sequential dependency – state transitions depend on robot actions $\mathcal{T} : (s_t, a_t) \rightarrow s_{t+1}$, robot actions depend on the sensor observations: $\pi : o_t \rightarrow a_t$, and observations depend on the state $\mathcal{O} : s_t \rightarrow o_t$. Thus, it ostensibly appears that physics and rendering outputs ($s_{t+1}$, $o_t$) cannot be computed in parallel from $s_t$ because computation of $a_t$ cannot begin till $o_t$ is available. We break this sequential dependency by changing the agent policy to be $\pi(a_t \,|\, o_{t-1})$ instead of $\pi(a_t \,|\, o_t)$. Thus, our agent predicts the current action $a_t$ not from the current observations $o_t$ but from an observation from 1 timestep ago $o_{t-1}$, essentially 'living in the past and acting in the future'.

This simple change means that we can generate $s_{t+1}$ on the CPU at the same time as $o_t$ is being generated on the GPU.

This strategy not only increases simulation throughput, but also offers two other fortuitous benefits – increased biological plausibility and improved sim2real transfer potential. The former is due to closer analogy to all sensors (biological or artificial) having a sensing latency (*e.g.*, the human visual system has approximately 150ms latency [61]). The latter is due to a line of prior work [62–64] showing that introducing this latency in simulators improves the transfer of learned agents to reality.

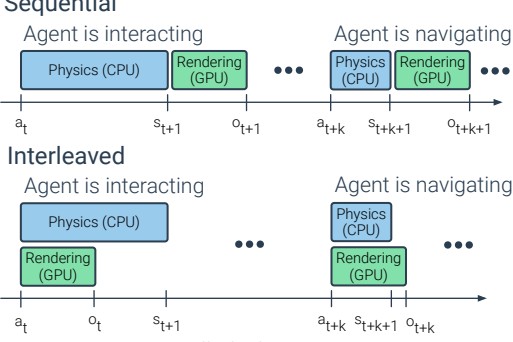

Figure 3: Interleaved physics and rendering. Top shows the normal sequential method of performing physics $(s_t, a_t) \rightarrow s_{t+1}$ then rendering $s_{t+1} \rightarrow o_{t+1}$. Bottom shows H2.0's interleaved physics and rendering.

### 4.3 Benchmarking

We benchmark using a Fetch robot, equipped with (up to) two RGB-D cameras ($128\times128$ pixels) in ReplicaCAD scenes under three scenarios: (1) Idle-1×RGB: with the robot initialized in the center of the living room somewhat far from furniture or any other object and taking random actions and equipped with a single RGB camera, (2) Idle-2×RGB-D, Idle-RGB with two RGB-D cameras, (3) Interact: with the robot initialized fairly close to the fridge and taking actions from a pre-computed trajectory that results in representative interaction with objects and equipped with two RGB-D cameras. Each simulation step consists of 1 rendering pass and 4 physics-steps, each simulating $1/120$ sec for a total of $1/30$ sec. New joint position goals are set every $1/30$ sec and a joint controller computes the joint torques to achieve the joint goals for the current joint state every $1/120$ sec. This is a fairly standard experimental configuration in robotics (with 30 FPS cameras and 120 Hz control). In this setting, a simulator operating at 30 steps per (wallclock) second (SPS) corresponds to 'real time'.

Benchmarking was done on machines with dual Intel Xeon Gold 6226R CPUs – 32 cores/64 threads (32C/64T) total – and 8 NVIDIA GeForce 2080 Ti GPUs. For single-GPU benchmarking processes are confined to 8C/16T of one CPU, simulating an 8C/16T single GPU workstation. For single-GPU multi-process benchmarking, 16 processes were used. For multi-GPU benchmarking, 64 processes were used with 8 processes assigned to each GPU. We used python-3.8 and gcc-9.3 for compiling H2.0. We report average SPS over 10 runs and a 95% confidence-interval computed via standard error of the mean. Note that 8 processes do not fully utilize a 2080 Ti and thus multi-process multi-GPU performance may be better on machines with more CPU cores.

| | 1 Process | | | | | | 1 GPU | | | | | | 8 GPUs | | | | | |
|---|---|---|---|---|---|---|---|---|---|---|---|---|---|---|---|---|---|---|
| | Idle 1×RGB | | Idle 2×RGB-D | | Interact 2×RGB-D | | Idle 1×RGB | | Idle 2×RGB-D | | Interact 2×RGB-D | | Idle 1×RGB | | Idle 2×RGB-D | | Interact 2×RGB-D | |
| H2.0 (Full) | 1191 | ±36 | 669 | ±13 | 510 | ±6 | 8186 | ±47 | 1926 | ±19 | 1660 | ±6 | 25734 | ±301 | 9542 | ±71 | 7699 | ±177 |
| - render opts. | 781 | ±9 | 364 | ±2 | 282 | ±2 | 6709 | ±89 | 1076 | ±6 | 1035 | ±3 | 18844 | ±285 | 6397 | ±43 | 5517 | ±31 |
| - physics opts. | 271 | ±3 | 252 | ±3 | 358 | ±6 | 2290 | ±5 | 1270 | ±30 | 1606 | ±6 | 7942 | ±50 | 5535 | ±41 | 6119 | ±51 |
| - all opts. | 242 | ±2 | 177 | ±3 | 224 | ±3 | 2223 | ±3 | 814 | ±2 | 941 | ±2 | 7192 | ±55 | 3965 | ±30 | 4829 | ±50 |

Table 2: Benchmarking H2.0 performance: simulation steps per second (higher better) over 10 runs and a 95% confidence-interval In Idle, the agent is executing random actions but not interacting with the scene, while Interact uses a precomputed trajectory and thus results in representative interaction with objects. To put these numbers into context, see Tab. 1. Reproduce these numbers at https://aihabitat.org/docs/habitat2.

Table 2 reports benchmarking numbers for H2.0. We make a few observations. The ablations for H2.0 (denoted by '- *render opts*', '-*physics opts*', and '-*all opts.*') show that principles followed in our system design lead to significant performance improvements.

Our 'Idle-1×RGB' setting is similar to the benchmarking setup of iGibson [36], which reports 100 SPS. In contrast, H2.0 single-process *with all optimizations turned off* is 240% faster (242 vs 100 SPS). H2.0 single-process with optimizations on is ∼1200% faster than iGibson (1191 vs 100 SPS). The comparison to iGibson is particularly illustrative since it uses the 'same' physics engine (PyBullet) as H2.0 (Bullet). We can clearly see the benefit of working with the low-level C++ Bullet rather than PyBullet and the deep integration between rendering and physics. However, we note the comparison between the two benchmarks is not exact since the robot type, number of objects, and

object assets are different. A direct comparison against other simulators is not feasible due to different capabilities, assets, hardware, and experimental settings. But a qualitative order-of-magnitude survey is illustrative – AI2-THOR [6] achieves 60/30 SPS in idle/interact, SAPIEN [34] achieves 200/400 SPS (personal communication), TDW [7] achieves 5 SPS in interact, and RLBench [35] achieves between 1 and 60 SPS depending on the sensor suite (personal communication). Finally, H2.0 scales well – achieving 8,186 SPS ($272\times$ real-time) multi-process on a single GPU and 25,734 SPS ($850\times$ real-time) on a single node with 8 GPUs. These $100\times$ simulation-speedups correspond to cutting experimentation time from 6-month cycle to under 2 days. iGibson also supports multi-process parallization for speeding simulation speeds beyond the reported single process numbers in the current version of the paper [36], and we recommend following updates of their work for more details.

## 5  The Pick Task: a Base Case of Rearrangement

We first carry out systematic analyses on a relatively simple robotic manipulation task: picking up one object from a cluttered 'receptacle'. This forms a 'base case' and an instructive starting point that we eventually expand to the more challenging Home Assistant Benchmark (HAB) (Sec. 6).

**Task Definition: `Pick` ($s^0$).** Fig. 4 illustrates an episode in the pick task. Our agent (a Fetch robot [1]) is spawned close to a receptacle (a table) that holds multiple objects (*e.g.* cracker box, bowl). The task for the robot is to pick up a target object with center-of-mass coordinates $s^0 \in R^3$ (provided in robot's coordinate system) as efficiently as possible without excessive collisions. We study systematic generalization to new clutter layout on the receptacle, to new objects, and to new receptacles. **Agent embodiment and sensing.** Fetch [1] is a wheeled base with a 7-DoF arm manipulator and a parallel-jaw gripper, equipped with two RGBD cameras (90° FoV, $128\times128$ pixels) mounted on its 'head' and arm.

It can sense its proprioceptive-state – arm joint angles (7-dim), end-effector position (3-dim), and base-egomotion (6-dim, also known as GPS+Compass in the navigation literature [3]). Note: the episodes in `Pick` are constructed such that the robot does not need to move its base. Thus, the egomotion sensor does not play a role in `Pick` but will be important in HAB tasks (Section 6).

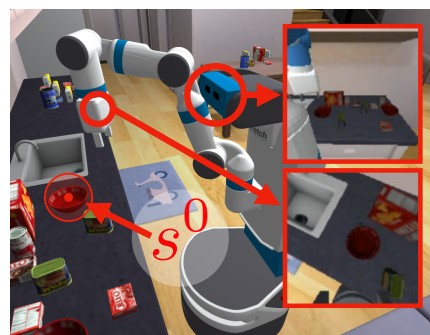

Figure 4: Fetch with head and arm cameras picking up a bowl from the counter.

**Action space: gross motor control.** The agent performs end-effector control at 30Hz. At every step, it outputs the desired *change* in end-effector position ($\delta x, \delta y, \delta z$); the desired end-effector position is fed into an inverse kinematics solver from PyBullet [37] to derive desired states for all joints, which are used to set the joint motor targets, achieved using PD control. The maximum end-effector displacement per step is 1.5cm, and the maximum impulse of the joint motors is 10Ns with a position gain of Kp=0.3. In `Pick`, the base is fixed but in HAB, the agent also emits linear and angular velocities for the base.

**Abstracted grasping.** The agent controls the gripper by emitting a scalar. If this scalar is positive and the gripper is not currently holding an object and the end-effector is within $15cm$ of an object, then the object closest to the end-effector is snapped into the parallel-jaw gripper. The grasping is perfect and objects do not slide out. If the scalar is negative and the gripper is currently holding an object, then the object currently held in the gripper is released and simulated as falling. In all other cases, nothing happens. For analysis of other action spaces see Appendix F.6.

**Evaluation.** An object is considered successfully picked if the arm returns to a known 'resting position' with the target object grasped. The agent fails if the accumulated contact force experienced by the arm/body exceeds a threshold of 5k Newtons. If the agent picks up the wrong object, the episode terminates. Once the object is grasped, the drop action is masked out meaning the agent will never release the object. The episode horizon is 200 steps.

**Methods.** We compare two methods representing two distinctive approaches to this problem: 1. **MonolithicRL**: a 'sensors-to-actions' policy trained end-to-end with reinforcement learning (RL). The visual input is encoded using a CNN, concatenated with embeddings of proprioceptive-sensing and goal coordinates, and fed to a recurrent actor-critic network, trained with DD-PPO [11] for 100

Million steps of experience (see Appendix C for details). This baseline translates our community's most-successful paradigm yet from navigation to manipulation.

2. **SensePlanAct** (**SPA**) pipeline: Sensing consists of constructing an accumulative 3D point-cloud of the scene from depth sensors, which is then used for collision queries. Motion planning is done using Bidirectional RRT [65] in the arm joint configuration space (see Appendix D). The controller was described in 'Action Space' above and is consistent with **MonolithicRL**. We also create **SensePlanAct-Priviledged** (**SPA-Priv**), that uses *privileged* information – perfect knowledge of scene geometry (from the simulator) and a perfect controller (arm is kinematically set to desired joint poses). The purpose of this baseline is to provide an upper-bound on the performance of **SPA**.

**Systematic Generalization.** With H2.0 we can compare how learning based systems generalize compared to **SPA** architectures. Tab. 3 shows the results of a systematic generalization study of 4 unseen objects, 3 unseen receptacles, and 20 unseen apartment layouts (from 1 unseen 'macro variation' in ReplicaCAD). In training the agent sees 9 objects from the YCB dataset kitchen and food categories (chef can, cracker box, sugar box, tomato soup can, tuna fish cap, pudding box, gelatin box, potted meat can, and

| Method | Seen | Unseen | | |
|---|---|---|---|---|
| | | Layouts | Objects | Receptacles |
| **MonolithicRL** | 91.7 $\pm 1.1$ | 86.3 $\pm 1.4$ | 74.7 $\pm 1.8$ | 52.7 $\pm 2.0$ |
| **SPA** | 70.2 $\pm 1.9$ | 72.7 $\pm 1.8$ | 72.7 $\pm 1.8$ | 60.3 $\pm 2.0$ |
| **SPA-Priv** | 77.0 $\pm 1.7$ | 80.0 $\pm 1.6$ | 79.2 $\pm 1.7$ | 60.7 $\pm 2.0$ |

Table 3: `Pick` generalization analysis: success rates with mean and standard error on 600 episodes (and across 3 seeds for **MonolithicRL**).

bowl). During evaluation it is tested on 4 unseen objects (apple, orange, mug, sponge). Likewise, the agent is trained on the counter, sink, light table, cabinet, fridge, dark table, and sofa receptacles (view in Fig. 11) but evaluated on the unseen receptacles of tv stand, shelves, and chair (view in Fig. 12).

**MonolithicRL** generalizes fairly well from seen to unseen layouts ($91.7 \rightarrow 86.3\%$), significantly outperforming **SPA** (72.7%) and even **SPA-Priv** (80.0%). However, generalization to new objects is challenging ($91.7 \rightarrow 74.7\%$) as a result of the new visual feature distribution and new object obstacles. Generalization to new receptacles is poor ($91.7 \rightarrow 52.7\%$). However, the performance drop of **SPA** (and qualitative results) suggest that the unseen receptacles (shelf, armchair, tv stand) may be objectively more difficult to pick up objects from since the shelf and armchair are tight constrained areas whereas the majority of the training receptacles, such as counters and tables, have no such constraints (see Fig. 12). We believe the performance of **MonolithicRL** will improve as more receptacles 3D assets become available since the training distribution was only 4 receptacles. We cannot make any such claims for **SPA**.

In the supplementary we also analyze different sensor input modalities (Appendix F.1), the surprising success of "blind" policies (Appendix F.2), the effect of different camera placements (Appendix F.3), different action spaces (Appendix F.6), the effect of the time delay on performance (Appendix F.5), and qualitative evidence of self-tracking (Appendix F.4).

# 6 Home Assistant Benchmark (HAB)

We now describe our benchmark of common household assistive robotic tasks. We stress that these tasks *illustrate* the capabilities of H2.0 but do not *delineate* them – a lot more is possible but not feasible to pack into a single coherent document with clear scientific takeaways.

**Task Definition.** We study three (families of) long-range tasks that correspond to common activities:

1. `TidyHouse`: Move 5 objects from random (unimpeded) locations back to where they belong (see Fig. 19a). This task requires no opening or closing and no objects are contained.
- Start: 5 target objects objects spawned in 6 possible receptacles (excluding fridge and drawer).
- Goal: Each target object is assigned a goal in a different receptacle than the starting receptacle.
- Task length: 5000 steps.

2. `PrepareGroceries`: Remove 2 objects from the fridge to the counters and place one object back in the fridge (see Fig. 19b). This task requires no opening or closing and no objects are contained.
- Start: 2 target objects in the fridge and one on the left counter. The fridge is fully opened.
- Goal: The goal for the target objects in the fridge are on the right counter and light table. The goal for the other target object is in the fridge.
- Task length: 4000 steps

3. `Set Table`: Get a bowl from a drawer, a fruit from fridge, place the fruit in the bowl on the table (see Fig. 19c).
   - Start: A target bowl object is in one of the drawers and a target fruit object in the middle fridge shelf. Both the fridge and drawer start closed.
   - Goal: The goal for the bowl is on the light table, the goal for the fruit is on top of the bowl. Both the fridge and drawer must be closed.
   - Task length: 4500 steps.

The list is in increasing order of complexity – from no interaction with containers (`TidyHouse`), to picking and placing from the fridge container (`PrepareGroceries`), to opening and closing containers (`Set Table`). Note that these descriptions are provided purely for human understanding; the robot operates entirely from a GeometricGoal specification [9] – given by the initial and desired 3D (center-of-mass) position of each target object $i$ to be moved $\left(s_i^0, s_i^*\right)_{i=1}^N$. Thus, `Pick` $\left(s_i^0\right)$ is a special case where $N = 1$ and $s_i^*$ is a constant (arm resting) location. For each task episode, we sample a ReplicaCAD layout with YCB [59] objects randomly placed on feasible placement regions (see procedural clutter generation in Section 3). Each task has 5 clutter objects per receptacle. Unless specified, objects are sampled from the 'food' and 'kitchen' YCB item categories in the YCB dataset.

The agent is evaluated on unseen layouts and configurations of objects, and so cannot simply memorize. We characterize task difficulty by the required number of rigid-body transitions (*e.g.*, picking up a bowl, opening a drawer). The task evaluation, agent embodiment, sensing, and action space remain unchanged from Section 5, with the addition of base control via velocity commands. Details on episode statistics, as well as the evaluation protocols are in Appendix G.

**Methods.** We extend the methods from Sec. 5 to better handle the above long-horizon tasks with a high-level STRIPS planner using a parameterized set of skills: `Pick`, `Place`, `Open fridge door`, `Close fridge door`, `Open drawer`, `Close drawer`, and `Navigate`. The full details of the planner implementation and how methods are extended are in Appendix H. Here, we provide a brief overview.

1. **MonolithicRL**: Essentially unchanged from Sec. 5, with the exception of accepting a list of start and goal coordinates $\left(s_i^0, s_i^*\right)_{i=1}^N$, as opposed to just $s_1^0$.

2. **TaskPlanning+SkillsRL** (**TP+SRL**): a hierarchical approach that assumes knowledge of a perfect task planner (implemented with STRIPS [27]) and the initial object containment needed by the task planner to break down a task into a sequence of parameterized skills: `Navigate`, `Pick`, `Place`, `Open fridge door`, `Close fridge door`, `Open drawer`, `Close drawer`. Each skill is functionally identical to **MonolithicRL** in Sec. 5 – taking as input a single 3D position, either $s_i^0$ or $s_i^*$. For instance, in the `Set Table` task, let $(a^0, a^*)$ and $(b^0, b^*)$ denote the start and goal positions of the apple and bowl, respectively. The task planner converts this task into:

$$\overbrace{\texttt{Navigate}(b^0), \texttt{Open drawer}(b^0)}^{\text{Open Drawer}}, \texttt{Pick}(b^0), \overbrace{\texttt{Navigate}(b^*), \texttt{Place}(b^*)}^{\text{Transport Bowl}}, \overbrace{\texttt{Navigate}(b^0), \texttt{Close drawer}(b^0)}^{\text{Close Drawer}},$$

$$\underbrace{\texttt{Navigate}(a^0), \texttt{Open fridge door}(a^0)}_{\text{Open Fridge}}, \underbrace{\texttt{Navigate}(a^*), \texttt{Place}(a^*)}_{\text{Transport Apple}}, \underbrace{\texttt{Navigate}(a^0), \texttt{Close fridge door}(a^0)}_{\text{Close Fridge}}.$$

Simply listing out this sequence highlights the challenging nature of these tasks.

3. **TaskPlanning+SensePlanAct** (**TP+SPA**): Same task planner as above, with each skill implemented via **SPA** from Sec. 5 except for `Navigate` where the same learned navigation policy from **TP+SPA** is used. **TP+SPA-Priv** is analogously defined. Crafting an **SPA** pipeline for opening/closing unknown articulated containers is an open unsolved problem in robotics – involving detecting and tracking articulation [66, 67] without models, constrained full-body planning [68–70] without hand engineering constraints, and designing controllers to handle continuous contact [71, 72] – making it out of scope for this work. Thus, we do not report **TP+SPA** on `Set Table`.

**Results and Findings.** Figure 5 shows progressive success rates for different methods on all tasks. Due to the difficulty of the full task, for analysis, the X-axis lists the sequence of agent-environment interactions (pick, place, open, close) required to accomplish the task, same as that used by the task-planner.[4] The number of interactions is a proxy for task difficulty and the plot is analogous to precision-recall curves (with the ideal curve being a straight line at 100%). Furthermore, since

---

[4]This sequence from the task plan is useful for experimental analysis and debugging, but does not represent the only way to solve the task and should be disposed in future once methods improve on the full task.

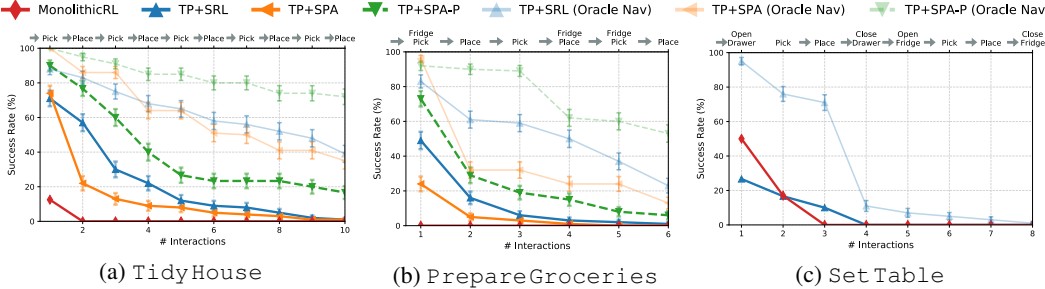

Figure 5: Success rates for Home Assistant Benchmark tasks. Due to the difficulty of full HAB tasks, we analyze performance as completing a part of the overall task. For the TP methods that use an explicit navigation skill, we indicate with an arrow in the interaction names where navigation occurs and include versions for learned and oracle navigation. Results are on unseen layouts with mean and standard error computed for 100 episodes.

navigation is often executed between successive skills, we include versions of the task planning methods with an oracle navigation skill. We make the following observations (See Appendix I for skill learning curves and **SPA** failure statistics):

1. **MonolithicRL** performs abysmally. We were able to train *individual* skills with RL to reasonable degrees of success (see Appendix I.2). However, crafting a *combined* reward function and learning scheme that elicits chaining of such skills for a long-horizon task, without any architectural inductive bias about the task structure, remained out of our reach despite prolonged effort.

2. Learning a navigation policy to chain together skills is challenging as illustrated by the performance drop between learned and oracle navigation. In navigation for the sake of navigation (PointNav [10]), the agent is provided coordinates of the reachable goal location. In navigation for manipulation (Navigate), the agent is provided coordinates of a target object's center-of-mass but needs to navigate to an unspecified non-unique *suitable* location from where the object is manipulable.

3. Compounding errors hurt performance of task planning methods. Even with the relatively easier skills in TidyHouse in Figure 5a all methods with oracle navigation gradually decrease in performance as the number of required interactions increases.

4. Sense-plan-act variants scale poorly to increasing task complexity. In the easiest setting, Tidy House with oracle navigation (Figure 5a), **TP+SPA** performs better than **TP+SRL**. However, this trend is reversed with learned navigation since **TP+SPA** methods, which rely on egocentric perception for planning, are not necessarily correctly positioned to sense the workspace. In the more complex task of PrepareGroceries (Figure 5b), **TP+SRL** outperforms **TP+SPA** both with and without oracle navigation due to the perception challenge of the tight and cluttered fridge. **TP+SPA** fails to find a goal configuration 3x more often and fails to find a plan in the allowed time 3x more often in PrepareGroceries than TidyHouse.

## 7 Societal Impacts, Limitations, and Conclusion

ReplicaCAD was modeled upon apartments in one country (USA). Different cultures and regions may have different layouts of furniture, types of furniture, and types of objects not represented in ReplicaCAD; and this lack of representation can have negative social implications for the assistants developed. While H2.0 is a fast simulator, we find that the performance of the overall simulation+training loop is bottlenecked by factors like synchronization of parallel environments and reloading of assets upon episode reset. An exciting and complementary future direction is holistically reorganizing the rendering+physics+RL interplay as studied by [73–78]. As illustrated in Figure 3, there is idle GPU time when rendering is faster than physics, because inference waits for both $o_t$ and $s_{t+1}$ to be ready despite not needing $s_{t+1}$. This is done because existing RL training systems expect the reward $r_t$ to be returned when the agent takes an action $a_t$, but $r_t$ is typically a function of $s_t$, $a_t$, and $s_{t+1}$. Reorganizing the rendering+physics+RL interplay is an exciting problem for future work.

We presented the ReplicaCAD dataset, the Habitat 2.0 platform and a home assistant benchmark. H2.0 is a fully interactive, high-performance 3D simulator that enables efficient experimentation involving embodied AI agents rearranging richly interactive 3D environments. Coupled with the ReplicaCAD data these improvements allow us to investigate the performance of RL policies against classical MP approaches for the suite of challenging rearrangement tasks we defined. We hope that the Habitat 2.0 platform will catalyze work on embodied AI for interactive environments.

## 8 Acknowledgements

Funding in direct support of this work: The Georgia Tech portion of this work supported by state funds from Georgia Tech (AS, ZK), NSF (DB), AFRL (DB), DARPA (DB), ONR YIPs (DB), ARO PECASE (DB), Amazon (DB). The SFU portion of this work is supported by a CIFAR AI Chair (AC), a Canada Research Chair (MS), and NSERC Discovery Grants (AC,MS). This work was also supported by FAIR (AC, EU, YZ, JT, NM, MM, DC, OM, AG, FM, WG) and Intel (VK).

Additional revenues related to this work: NSF (ZK), DARPA (ZK), ONR (ZK), NGA (ZK), Samsung (ZK), Airbus (ZK), consulting for Marble Inc (ZK), paid talk by Data Science Connect (ZK), sponsored research funding by FAIR (AC,MS).

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
