# A  ReplicaCAD Further Details

## A.1  Dataset Details

The 20 micro-variations of the 5 macro-variations of the scene were created with the rule of swapping at least two furniture pieces and perturbing the positions of a subset of the other furniture pieces. The occurrences of various furniture objects in these 100 micro-variations are illustrated in Fig. 6. Several furniture objects such as 'Beanbag' and 'Chair' occur more frequently with multiple instances in a some scenes while others such as 'Table 03' occur less frequently.

We also analyze the object categories of all objects in the original 6 'FRL-apartment' space recreations. We map each of the 92 objects to a semantic category and list the counts per semantic category in a histogram in Fig. 7. Since these spaces have a large kitchen area, there is a larger ratio of kitchen objects such as 'Kitchen utensil' and 'Bowl'.

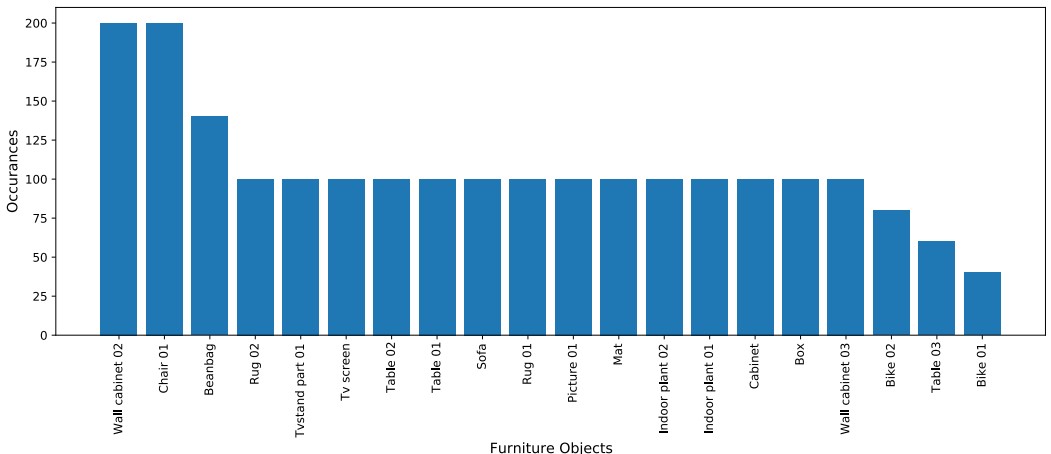

Figure 6: Number of occurrences for each furniture type across the 100 micro-variations out of the total 111 ReplicaCAD scenes.

Top down views of the 5 'macro variations' of the scenes are shown in Fig. 8. These variations are 5 semantically plausible configurations of furniture in the space generated by a 3D artist. Each surface is annotated with a bounding box, enabling procedural placement of objects on the surfaces. For each of these 5 variations, we generate 20 additional variations, giving 105 scene layouts.

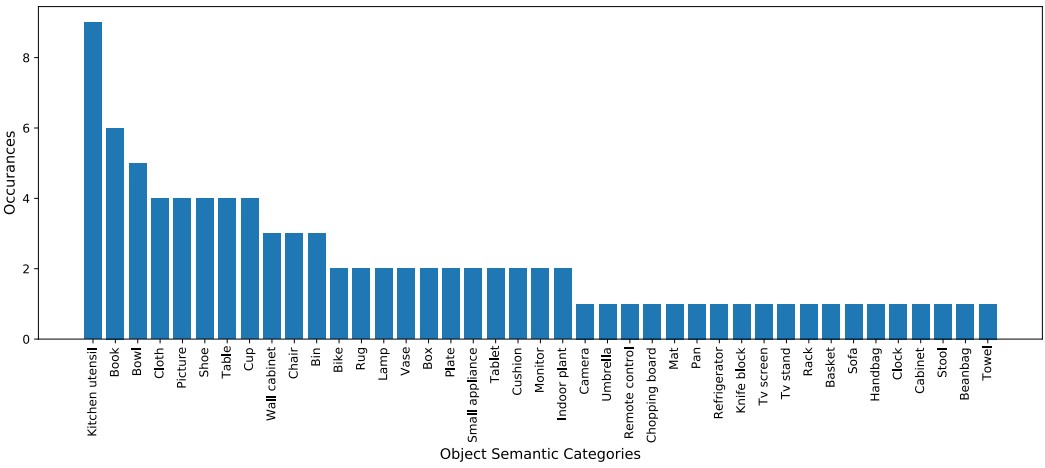

Figure 7: Histogram of objects belonging to each semantic category out of the 92 overall objects.

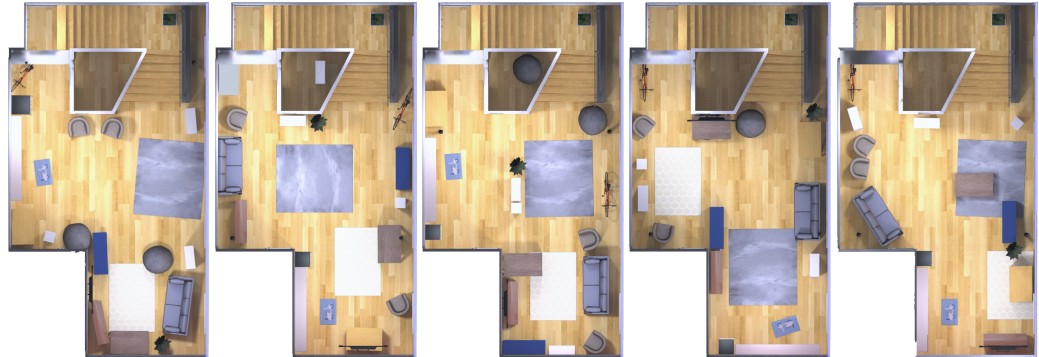

Figure 8: The 5 ReplicaCAD 'macro variations' of semantically plausible configurations of furniture in the apartment space. Objects are procedurally added on furniture and surfaces using the annotated supporting surface and containment volume information provided by ReplicaCAD.

## A.2 Further Comparison to Prior Work

We further situate ReplicaCAD with respect to prior work. ReplicaCAD is one house with 111 layouts (6 from real scans, 105 artist created), The TDW Transporter Challenge [7] has 15 houses, ManipulaTHOR [33] has 30 rooms, RoomR from Visual Room Rearrangement [58] has 120 rooms, iGibson [36] has 15 houses, and VirtualHome [51] has 6 houses. While house sizes are not reported for these datasets, ReplicaCAD includes kitchens and living rooms, ManipulaTHOR includes kitchens, and RoomR, iGibson, VirtualHome, and TDW include kitchens, living rooms, bedrooms, and bathrooms. ManipulaTHOR and RoomR have single room, not house scale, scenes. In terms of object assets, ReplicaCAD has 92 objects, TDW Transporter challenge has 50 objects, ManipulaTHOR has 150 objects, RoomR has 118 objects, iGibson has 57 objects, Virtual Home has 357 objects, and the Sapien PartNet dataset has 2,346 objects.

## B Habitat 2.0 (H2.0): Features in Detail

### B.1 Simplify and reuse

Scenes with many interactive objects can pose a challenge for limited GPU memory. To mitigate this, we apply GPU texture compression (the Basis 'supercompressed' format [79]) to all our 3D assets, leading to 4x to 6x (depending on the texture) reduction in GPU memory footprint. This allows more objects and more concurrent simulators to fit on one GPU and reduces asset import times. Another source of slowdown are 'scene resets' – specifically, the re-loading of objects into memory as training/testing loops over different scenes. We mitigate this by pre-fetching object assets and caching them in memory, which can be quickly *instanced* when required by a scene, thus reducing the time taken by simulator resets. Finally, computing collisions between robots and the surrounding geometry is expensive. We create convex decompositions of the objects and separate these simplified collision meshes from the high-quality visual assets used for rendering. We also allow the user to specify simplified collision geometries such as bounding boxes, and per-part or merged convex hull geometry. Overall, this pipeline requires minimal work from the end user. A user specifies a set of objects, they are automatically compressed in GPU memory, cached for future prefetches, and convex decompositions of the object geometry are computed for fast collision calculations.

### B.2 Motion Planning Integration

Finally, H2.0 includes an integration with the popular Open Motion Planning Library (OMPL), giving access to a suite of motion planning algorithms [80]. This enables easy comparison against classical sense-plan-act approaches [26]. These baselines are described in Sec. 5 with details in Appendix D.

### B.3 Future Simulator Work

Our solution also reveals the natural next problem of how to interleave physics, rendering, *and* policy inference. As illustrated in Fig. 3, there is idle GPU time when rendering is faster than physics,

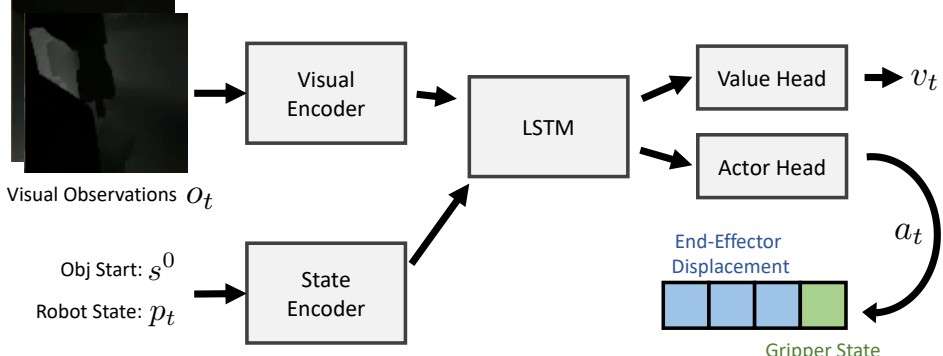

Figure 9: **Learned (Mono)** policy architecture. The policy maps egocentric visual observations $o_t$, the task-specification in the form of a geometric object goal $s^0$, and the robot proprioceptive state $p_t$ into an action $a_t$ which controls the arm and gripper. A value output is also learned for the PPO update.

because inference waits for both $o_t$ and $s_{t+1}$ to be ready despite not needing $s_{t+1}$. This is done to maintain compatibility with existing RL training systems, which expect the reward $r_t$ to be returned when the agent takes an action $a_t$, but $r_t$ is typically a function of $s_t$, $a_t$, and $s_{t+1}$. Holistically reorganizing the rendering+physics+RL interplay is an exciting open problem for future work.

## C  MonolithicRL Details

### C.1  Architecture

The **MonolithicRL** architecture consists of a visual encoder which takes as input the egocentric visual observation and a state encoder neural network which takes as input the object start position and the current proprioceptive robot state. Both the image and state inputs are normalized using a per-channel moving average. $RGB$ and $D$ input modalities are fused by stacking them on top of each other. These two encodings are passed into an LSTM module which are then processed by an actor head to produce the end-effector and gripper state actions and a value head to produce a value estimate. The agent architecture is illustrated in Fig. 9.

### C.2  Training

The agent is trained with the following reward function

$$r_t = 20\mathbb{I}_{success} + 5\mathbb{I}_{pickup} + 20\Delta^o_{arm}\mathbb{I}_{!holding} + 20\Delta^r_{arm}\mathbb{I}_{holding} - \max(0.001C_t, 1.0)$$

Where $\mathbb{I}_{holding}$ is the indicator if the robot is holding an object, $\mathbb{I}_{success}$ is the indicator for success, $\mathbb{I}_{pickup}$ is the indicator if the agent just picked up the object, $\Delta^o_{arm}$ is the change in Euclidean distance between the end-effector and target object (if $d_t$ is the distance between the two at timestep $t$, then $\Delta^o_{arm} = d_{t-1} - d_t$), and $\Delta^r_{arm}$ is the change in distance between the arm and arm resting position. $C_t$ is the collision force in Newtons at time $t$.

We train using the DDPPO algorithm [11] with 16 concurrent processes per GPU across 4 GPUs for 64 processes in total with a preemption threshold of 60%. For the PPO [81] hyperparameters, we use a value loss coefficient of 0.1, entropy loss coefficient of 0.0001, 2 mini-batches, 2 epochs over the data per update, and a clipping parameter of 0.2 We use the Adam [82] with a learning rate of 0.0001. We also clip gradient norms above a magnitude of 0.5. We train for 100M steps of experience and linearly decay the learning rate over the course of training. We train on machines using the following NVIDIA GPUs: Titan Xp, 2080 Ti, RTX 6000.

## D  Motion Planning

In this section, we provide details on our motion planning based sub-task policies that can be composed together to solve the overall task analogous to the Learned policy. These approaches

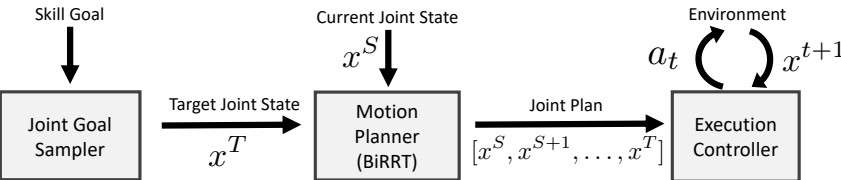

Figure 10: The three stages of our robotics pipeline for **SPA-Priv** and **SPA**. Starting from a high-level objective such as picking a certain object, the "Joint Goal Sampler" produces the necessary goal for the motion planner to plan to based on random sampling and inverse-kinematics. The motion planner then plans a path in joint space from the current joint angles to the desired joint angles. The executor then translates the motion planner into torque actions for the robot motors.

employ a more traditional non-learning based robotics pipeline [83]. Our pipeline consists of three stages: joint goal sampling, motion planning, and execution as illustrated in Figure 10.

We exclusively use the sampling-based algorithm RRTConnect [84] (bidirectional rapidly-exploring random tree) as the motion planner given that it is one of the state-of-the-art methods that the robotics literature frequently builds on and compares to [85–93] and for which a well maintained open source implementation is available in the OMPL library [80] (open motion planning library). Since it does not employ learning, it also serves as a stand-in for a more traditional non-learning based robotics pipeline.

Our aim with the current baselines is to demonstrate a strong starting point and our hope is that it drives adoption within the robotics community to develop and benchmark their algorithms, learning based or otherwise on this platform. For instance, work in the area of motion planning has made several advancements with new sampling techniques [94, 95] and optimization based methods [86, 87, 89, 96], but largely operated on the assumption of a reliable perception stack. However, difficulty in obtaining maintained open source implementations that are not tied to a specific hardware or have complex dependencies like ROS [97] have also posed challenges in bringing the vision and robotics communities together under a common set of tasks. More recent work has however begun utilizing learning and transitioning towards hybrid methods, for example learning distributions for sampling [90], using reinforcement [98] or differentiating through the optimization [99].

We implement two variants that defer in how they handle perception: one that uses privileged information from the simulator (**SPA-Priv**) and one that uses egocentric sensor observations (**SPA**). **SPA** uses depth sensor to obtain a 3D point cloud in the workspace of the robot at the measurement instance which is used for collision checking. Since the arm can get in the way of the depth measurement, the arm optionally lowers so the head camera on the Fetch robot can sense the entire workspace. If it is not possible to lower the camera (as in the case of holding an object), the detected points consisting of the robot's arm are filtered out and detected points from prior robot positions and orientations are accumulated (which is possible since we have perfect localization). **SPA-Priv** on the other hand can directly access the ground truth scene geometry for collision checking. **SPA-Priv** plans in an identical Habitat simulator instance as the current scene by directly setting the state and checking for collisions using the duplicate Habitat simulator instance. When the robot is holding an object, **SPA-Priv** updates the position of the held object based on the current joint states for collision checking in planning. The full, not simplified robot model is used for collision checking.

A wrapper exposes a Habitat or PyBullet simulator instance to OMPL to perform the motion planning. Specifically, this exposes a check for collision based on a set of Fetch robot arm joint angles. Sampling is constrained to the valid joint angles of the Fetch robot.

Motion planning is used as a component in performing skills. At a high-level many skills repeat the same steps. First, determine the specific goal as a target joint state of the robot arm for the planner based off the desired interaction of the skill. This could be a grasp point for picking an object up, a valid position of the arm to drop an object, a position of the arm which can grasp the handle, etc. A combination of IK, random sampling and collision checks, are used to solve this step. Next, a planning algorithm from OMPL is invoked to find a valid sequence of arm joint states to achieve the goal. Finally, the robot executes the plan by consecutively setting joint motor position targets based on the planned joint positions.

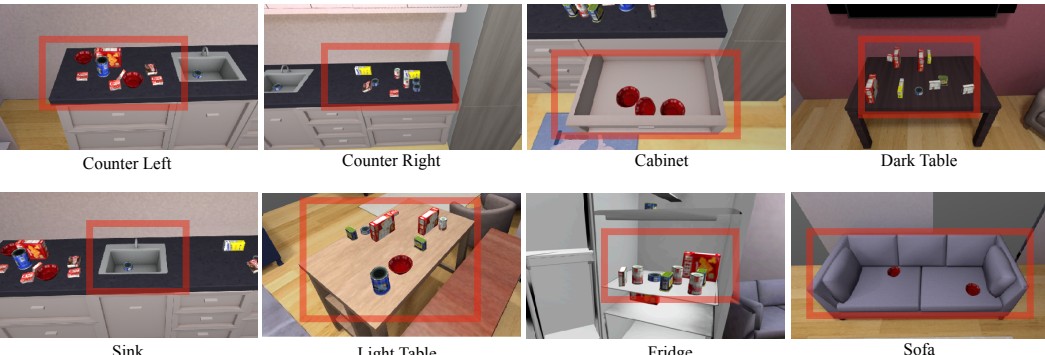

Figure 11: Receptacles used in Pick task training. One policy is trained to pick target objects across all receptacles. Some receptacles such as the Fridge, Sink, and Cabinet are more challenging due to the tight spaces and obstacle geometry.

- `Pick`: First sample a grasp position on the object. In the case of **SPA-Priv**, use the bounding box of the object to sample grasp points. The grasp points are the the centers of the box faces. For each center, find a valid collision free arm position which has its gripper at the box center by using IK starting from 1,000 randomly sampled arm positions. Take the solution which had the closest distance to the arm resting position in joint space. For the case of **SPA**, compute a best fit bounding box around the points where the desired object is and repeat the same procedure. This grasp planning produces a desired arm joint state, now use BiRRT to solve the planning problem. After the robot picks the object, it plans a path back to the arm resting position, using the stored joint states of the resting arm as the target.

- `Place`: The same as Pick, but now sample a goal position as a joint state which has the object at the target. For **SPA-Priv**, this uses the exact object model. For **SPA** this uses a heuristic distance of the gripper to the desired object placement.

RRTConnect is used to find the joint space path of the robot arm starting from the current joint position to a goal joint position that is calculated with inverse kinematics given some desired 2D or 3D end effector position. The specific sub-tasks are implemented as follows. This is also possible to accomplish with a single planning call using CBiRRT [100], a variant of RRTConnect that supports task space constraints, but is currently not available within OMPL.

We use a 30 second timeout for the planning. A step size of 0.1 radians is used for the step size in the RRTConnect algorithm. All planning is run on a machine using a Intel(R) Core(TM) i9-9900X CPU @ 3.50GHz.

# E   Pick Task: A Base Case of Rearrangment Details

## E.1   Generalization Experiments Details

We evaluate generalization of the Pick skill to unseen scenes, objects types, and receptacles. In training the agent sees 9 objects from the kitchen and food categories of the YCB dataset (chef can, cracker box, sugar box, tomato soup can, tuna fish cap, pudding box, gelatin box, potted meat can, and bowl). During evaluation it is tested on 4 unseen objects (apple, orange, mug, sponge). Likewise, the agent is trained on the counter, sink, light table, cabinet, fridge, TV stand, and sofa receptacles (visualized in Fig. 11) but evaluated on the unseen receptacles of dark table, shelves, and chair (visualized in Fig. 12).

## E.2   Evaluation

The agent fails if the accumulated contact force experienced by the arm/boduy exceeds a threshold of 5k Newtons. The agent must return to a resting end-effector position while holding the target object. If the agent picks up the wrong object, the episode terminates. Once the object is grasped, the drop

action is masked out meaning the agent will never release the object. The episode horizon is 200 steps.

### E.3 Agent

The agent controls the arm with a 4D continuous vector. The first three dimensions control the relative displacement of the end-effector $(\delta x, \delta y, \delta z)$. For end-effector control we use the IK library in PyBullet. Future work on H2.0 will port the PyBullet implementation of IK to be tightly coupled with the robotics API in H2.0. At every step, the target end-effector location is updated by a relative end-effector displacement, and then IK solves for the target joint states which are then used to set the joint motor targets for the arm. The maximum end-effector displacement at every step is 1.5cm. The maximum impulse of the joint motors on the Fetch arm are 10Ns with a position gain of Kp=0.3.

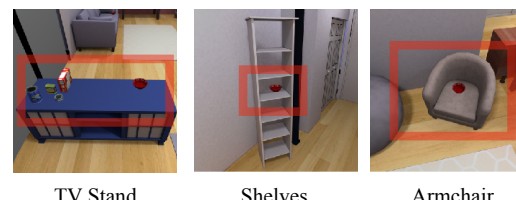

TV Stand      Shelves      Armchair

Figure 12: The agent is evaluated on the three unseen receptacles above. These receptacles were hand chosen to test diverse solutions. The chair requires avoiding the side arms, the shelf requires picking from a confined shelving space from a side angle, and finally the table is visually different from those in training.

The gripper is controlled through the final single continuous number. If this number is greater than or equal to 0, the agent invokes the discrete grasp which snaps the closest object based on the object center of mass to the robot robot end-effector if the object is within $15cm$. If the robot is already holding an object when the discrete grasp is executed, nothing happens. If the gripper value is less than 0, the agent releases the currently held object. If the robot is not holding any object, nothing happens.

## F  Pick Task Further Analysis Experiments

### F.1  Sensor Analysis for MonolithicRL: Blind agents learn to `Pick`

We also use H2.0 to analyze sensor trade-offs at scale (70M steps of training). We use the training and evaluation setting from Sec. 5.

Figure 13 shows success rates on unseen layouts, but seen receptacles and objects types, vs training steps of experience for **MonolithicRL** equipped with different combinations of sensors {Camera $RGB$, Depth $D$, proprioceptive-state $ps$}. To properly handle sensor modality fusions, we normalize the image and state inputs using a per-channel moving average. We note a few key findings:

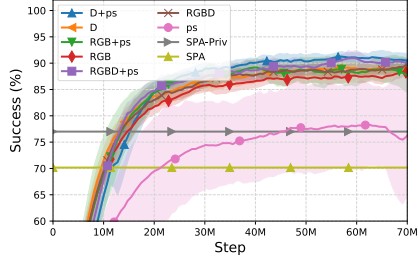

Figure 13: **MonolithicRL** sensor ablations: Success rates on unseen layouts ($N$=500) vs training steps. Mean and std-dev over 3 training runs.

1. Variations of $RGB$ and $D$ all perform similarly, but $D+ps$ slightly performs marginally better ($\sim 0.5\%$ over $RGBD+ps$ and $\sim 2\%$ over $RGB+ps$). This is consistent with findings in the navigation literature [12] and fortuitous since depth sensors are faster to render than $RGB$.

2. Blind policies, *i.e.* operating entirely from proprioceptive sensing are *highly* effective (78% success). This is surprising because for unseen layouts, the agent has no way to 'see' the clutter; thus, we would expect it to collide with the clutter and trigger failure conditions. Instead, we find that the agent learns to 'feel its way' towards the goal while moving the arm slowly so as to not incur heavy collision forces. Quantitatively, blind policies exceed the force threshold 2x more than sighted ones and pick the wrong object 3x more. We analyze this hypothesis further in Appendix F.2.

### F.2  Blind Policy Analysis

To further investigate the hypothesis that the blind 'feels its way' to the goal, we analyze how *efficient* the two are at picking up objects, using the Success weighted by Completion Time (SCT)

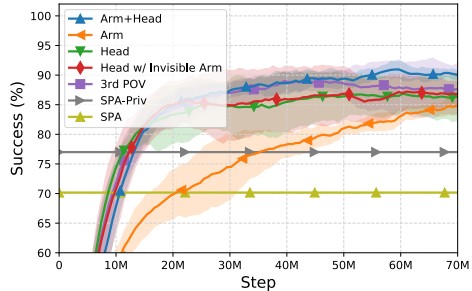

Figure 15: Camera placement analysis: Success rates on unseen layouts ($N$=500) vs training steps. Mean and std-dev over 3 training runs.

metric [101]. Specifically, $SCT = \text{Success} \cdot \left(\text{time taken by agent}/\text{time taken by oracle}\right)$. We use an upper-bound on the oracle-time: $2*\text{Euclidean distance(end-effector, goal)}/\text{maximum speed of end-effector}$. For ease of analysis, we use a simplified Pick setting with only the 'left counter' receptacle. The robot starts in front of the counter receptacle facing the wall. $N(0, 50)$cm is added to both the $x, y$ position of the base, $\mathcal{N}(0, 0.15)$ radians is added to the base orientation, $\mathcal{N}(0, 5)$cm is added to the $x, y, z$ of the starting end-effector position. 5 objects are randomly placed on the counter from the 'food' or 'kitchen' item categories of the YCB dataset. One of the objects is randomly selected to be the target object.

Figure 14 shows the SCT (on unseen layouts) as a function of the collision-force threshold used during training for policies trained for 100M steps. We find that sighted policies (Depth) are remarkably efficient across the board, achieving over $80\%$ SCT. Since we use a crude upper-bound on the oracle time it is unclear if a greater SCT is possible. The sighted policies may be discovering nearly maximally efficient trajectories, which would be consistent with known results in navigation [11]. The collision threshold is not related to performance, since the collision threshold is also used in training and will affect training. Very low collision thresholds result in conservative policies which avoid any hard collisions with objects and succeed more. Blind policies are significantly less efficient and *improve* in efficiency as the allowed collision threshold is reduced

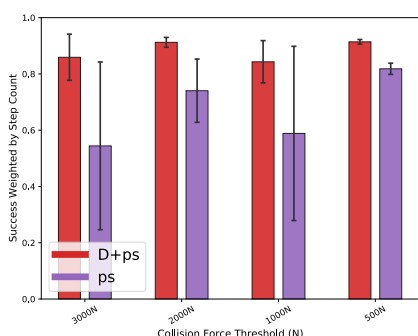

Figure 14: Path efficiency for sighted vs blind policies vs amount of collision allowed ($N$=3).

### F.3 Camera Placement: Arm cameras are most useful; Suggestive evidence for self-tracking

One advantage of fast simulation is that it lets us study robot designs that may be expensive or even *impossible* to construct in hardware. We use the same experimental settings as Sec. F.1, training the policies to pick objects from 8 receptacles (receptacles depicted in Fig. 11). 'Arm' and 'Head' placements were already described in Sec. 5. '3rdPoV' is a physically-implausible camera placement with a view from over the robot's shoulder (commonly used in video games and image-based RL papers *e.g.* [102]). 'Invisible Arm' is a physically-impossible setting where the robot's arm is physically present and interacts with the scene but is not visible in the cameras.

Fig. 15 shows performance on unseen layouts (vs training steps) for different camera placements on Fetch. While all camera placements perform generally well (80-90% success), The combination of head and arm camera performs best (at $92\%$ success). The arm only camera performs the worst, being slower to learn and only ultimately achieving $85\%$ success rate.

### F.4 Emergence of Self-Tracking

In order to qualitatively analyze the performance of the Pick policies, we visually interpret the saliency of the trained policy via Grad-CAM maps [103] computed with respect to the actions of the robot. To generate these Grad-CAM heatmaps, we follow the protocol laid down in Grad-CAM [103]

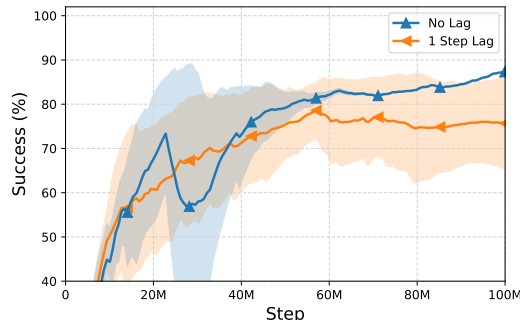

Figure 17: Effect of the time-delay on performance on the picking skill. Averages and standard deviations across 3 seeds. 1-step has high-variance results which could be reduced with more seeds.

and compute the gradient of each of the four continuous action values ('displacement' in three directions and 'grab') with respect to the activations of the final convolutional layer of the visual encoder. Subsequently, we average the heatmaps for each of the 'displacement' actions to give us an overall sense of saliency for the robot's displacement based on the input at each step, and perform the required post-processing to overlay this on top of the input frame. Fig. 16 shows the overall displacement maps for the robot in three different scenes and demonstrates the emergence of self-tracking behavior. In different scenes from cameras mounted on the Head as well as the Arm, we find a consistent trend that the maps highlight arm joints suggesting that the agent has learned to track the arm.

Caveat: we stress that saliency maps and the act of drawing inferences from them are fraught with a host of problems (see [104, 105] for excellent discussions). This analysis should be considered a speculative starting point for further investigation and not a finding it itself.

### F.5 Effect of Time-Delay on Performance

We studied the effect the time delay in Fig. 17 in the same experimental setting as Appendix F.2 and find that the time delay has a minimal impact on performance. The 1-step delayed time has large variance which could be reduced through more seeds.

### F.6 Action Space Analysis

Action spaces other than end-effector control are possible in H2.0. We compare end-effector versus velocity control in the Pick skill in Figure 18 in the same experimental setting as Appendix F.2. For velocity control, the policy outputs a 7 dimension vector representing the relative displacement of the position target for the PD controller. Despite, this higher dimension action space, velocity control learns just as well as end-effector control for the picking skill.

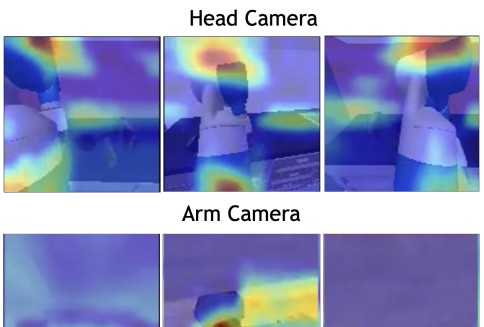

Figure 16: Grad-CAM saliency maps for three different scenes from cameras mounted on the Head and the Arm. Notice that the arm-joints are considered particularly salient in both cases across scenes.

## G Home Assistant Benchmark Experimental Setup Details

### G.1 Evaluation

For each task, 100 evaluation episodes are generated. These evaluation episodes have unseen micro-variations of the furniture not seen during any training for the learned methods. Object positions are

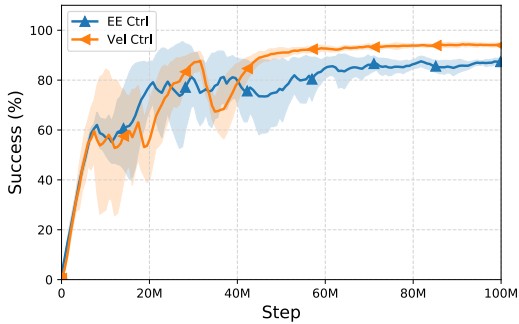

Figure 18: Comparison of end-effector and velocity control for the picking skill. Averages and standard deviations across 3 seeds. Both end-effector and velocity control are able to solve the task.

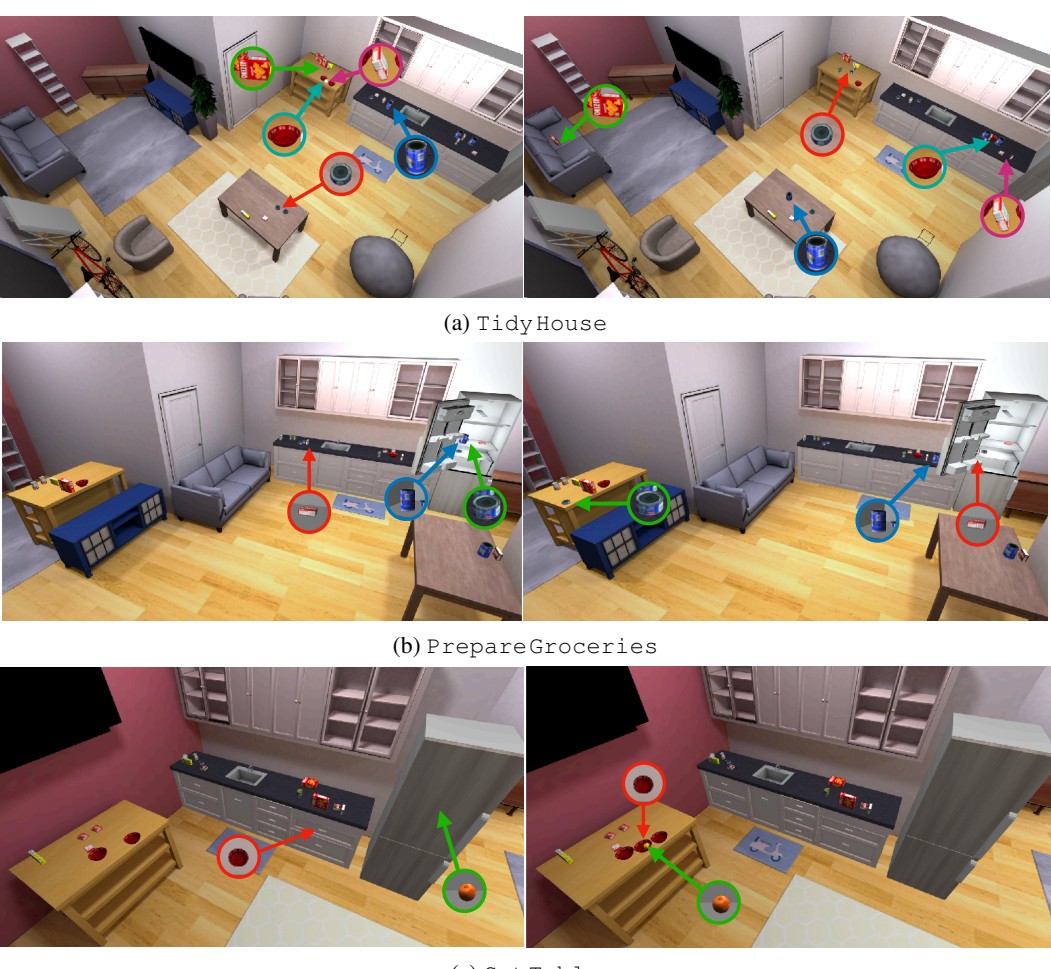

(a) `TidyHouse`

(b) `PrepareGroceries`

(c) `SetTable`

Figure 19: Example start and goal state for `TidyHouse`, `PrepareGroceries`, and `SetTable`. Left column: example starting state for tasks, right column: associated goal state color coded by object. Inset images and arrows denote the object start or goal position. Objects in `SetTable` start in the closed drawer and fridge.

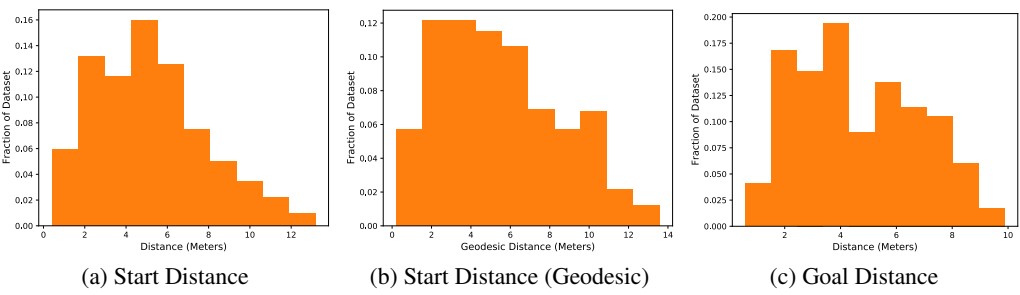

(a) Start Distance  (b) Start Distance (Geodesic)  (c) Goal Distance

Figure 20: `TidyHouse` Rearrangement dataset statistics.

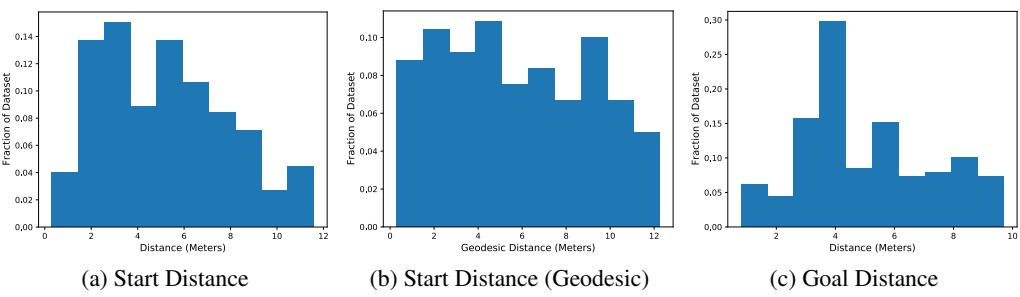

(a) Start Distance  (b) Start Distance (Geodesic)  (c) Goal Distance

Figure 21: `SetTable` Rearrangement dataset statistics.

randomized between episodes and the robot spawns at a random position in the scene. See Figures 20 to 22 for rearrangement dataset statistics for the Home Assistant Benchmark task definitions.

For each task, success is evaluated based on if all target objects were placed within 15cm of the goal position for that object, object orientation is not considered. To make evaluation easier, there was no collision threshold applied for full task evaluation.

### G.2 Partial Evaluation

Since our tasks are very challenging, we also feature partial evaluation of the tasks up to only a part of the overall rearrangements needed to solve the task. These partial task solving rearrangements are listed below, note each rearrangement builds upon the previous rearrangements and the robot must complete each of the previous rearrangements as well.

- `TidyHouse`: (1) pick object 1, (2) place object 1, (3) pick object 2, etc. Each of the 10 interactions is picking and placing a successive target object.
- `PrepareGroceries`: (1) pick first fridge object, (2) place first fridge object on counter, (3) pick second fridge object, (4) place second fridge object on table, (5) pick counter object, (6) place counter object in fridge. Like `TidyHouse`, each of the interactions is picking and placing an object.
- `SetTable`: (1) open the drawer, (2) pick the bowl from the drawer, (3) place the bowl on the table, (4) close the drawer, (5) open the fridge, (6) pick the apple from the fridge, (7) place the apple in the bowl, (8) close the fridge.

## H Home Assistant Benchmark Baseline Method Details

### H.1 Planner Details

All three of the hierarchical methods, **TP+SRL**, **SPA**, and **SPA-Priv** utilize a STRIPS high-level planner. A PDDL style domain file defines a set of predicates and actions. We define the following predicates

- *in(X,Y)*: Is object $X$ in container $Y$?
- *holding(X)*: Is the robot holding object $X$?

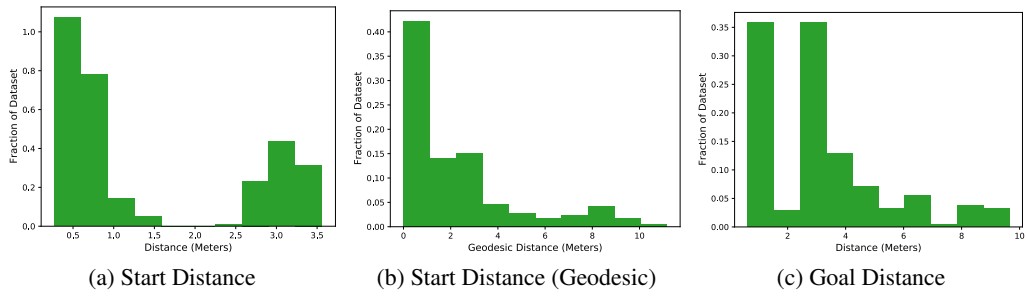

(a) Start Distance        (b) Start Distance (Geodesic)        (c) Goal Distance

Figure 22: `PrepareGroceries` Rearrangement dataset statistics.

- *at(X,Y)*: Is entity $X$ within interacting distance of $Y$?
- *is_closed(X)*: Is articulated object $X$ in the closed state (separately defined for each articulated object)?
- *is_open(X)*: Is articulated object $X$ in the open state?

And the following actions where each action is also linked to an underlying skill.

- *pick(X)*: Pick object X (Figure 23a):
  - Precondition: *at(robot, X)*. We also include the precondition *is_open(Z)* if *in(X, Z)* is true in the starting set of predicates.
  - Postcondition: *holding(X)*
  - Skill: Pick
- *place(X, Y)*: Place object X at location Y (Figure 23b):
  - Precondition: *at(robot,Y), holding(X)*. We also include the precondition *is_open(Z)* if *in(X, Z)* is true in the starting set of predicates.
  - Postcondition: *!holding(X),at(X,Y)*
  - Skill: Place
- *open(X)*: Open articulated object X (Figures 23c and 23e):
  - Precondition: *at(robot, X), is_closed(X), !holding(Z), ∀ Z*
  - Postcondition: *is_open(X)*
  - Skill: If X is the fridge entity, then `Open fridge door`, if X is the drawer entity then `Open drawer`.
- *close(X)*: Close articulated object X (Figures 23d and 23f):
  - Precondition: *at(robot, X), is_open(X), !holding(Z), ∀ Z*
  - Postcondition: *is_closed(X)*
  - Skill: If X is the fridge entity, then `Close fridge door`, if X is the drawer entity then `Close drawer`.

Each task defines the initial set of predicates and the goal set of predicates. We use a STRIPS planner to find a set of actions to transform the starting predicates into the goal predicates. Since we only deal with object rearrangement problems, the goal predicates of each task are of the form *at(obj_X, obj_goal_X)* for each object $X$ to be rearranged. `TidyHouse` and `SetTable` includes *is_closed(fridge),is_closed(drawer)* into the goal and starting predicates while `PrepareGroceries` includes *is_open(fridge),is_open(drawer)* into the goal and starting predicates. The starting predicates which specify containment and are listed below:

- `SetTable`: *in(bowl,drawer),in(fruit,fridge),in(fruit_goal,bowl_goal)*
- `PrepareGroceries`: No containement specified for this task (everything starts open).
- `TidyHouse`: No containement in this task.

We run the STRIPS planner once per task and save the minimum length solution. The saved plan is used as the sequence of agent-environment interactions for partial evaluation in Section 6.

## H.2   RL Skill Training

Each skill is trained to perform a desired interaction. To facilitate better transitions between skills, skills must reset the robot arm to a "resting position" with the end-effector at a certain position in front of the robot. Since the agent has access to proprioceptive state, this also serves as a termination signal for the skill.

For all skills $\Delta^o_{arm}$ is the change in distance between the end-effector and object (if $d_t$ is the distance between the two at timestep $t$ then $\Delta^o_{arm} = d_{t-1} - d_t$). $\Delta^r_{arm}$ is the change in distance between the arm end-effector and resting position, $\Delta^g_o$ is the change in distance between the object and object goal, $\Delta^h_{arm}$ is the change in distance between the end-effector and articulated object interaction point, and $\Delta^g_a$ is the distance between the articulated object state and its goal state. Skills incorporate a collision penalty with $C_t$, the collision force in Newtons at time $t$. Episodes terminate if they exceed a collision threshold $C_{max}$ in Newtons. By default in each skill training episode the robot base angle varies by 20 degrees, base $x, y$ position by 0.1 meters, and end-effector $x, y, z$ by 0.05 meters. For each skill, the maximum episode length is 200 steps.

Arm control refers a 3D relative offset of the end-effector and a 1D gripper value. Base control is a 2D linear and angular velocity. For all rewards, implicitly for time $t$, $\mathbb{I}_{holding}$ is the indicator for if the robot is holding an object, $\mathbb{I}_{force}$ is the indicator for if the force collision threshold was exceeded.

For training each skill, we utilize 5,000 training configurations. The full task where these skills are deployed are in unseen scene configurations and unseen object placements. We also show evaluation for each skill on an evaluation set of 500 configurations in Appendix I.2.

- Pick $(s^0_i)$ Pick the object at starting state $s^0_i$:
  - Starting state: Objects and clutter is randomly spawned on one of 6 receptacles (sofa, fridge, counter left, counter right, light wood table, dark wood table). Robot is facing the object with default noise applied to the base, orientation, and end-effector.
  - Success: Robot holding the target object and in the resting position.
  - Failure: $C_{max} = 5000$. The episode also terminates if the robot picks up the wrong object.
  - Reward: $\mathbb{I}_{wrong}$ indicates the policy picked up the wrong object.

$$r_t = 20\mathbb{I}_{success} + 5\mathbb{I}_{pickup} + 20\Delta^o_{arm}\mathbb{I}_{!holding} + 20\Delta^r_{arm}\mathbb{I}_{holding} - \max(0.001C_t, 1.0)$$
$$-10\mathbb{I}_{force} - 5\mathbb{I}_{wrong} - 5\mathbb{I}_{dropped}$$

  - Agent action space: Arm control. Once an object is picked, the gripper scalar action is masked out until the skill terminates to prevent dropping the object.
  - Agent observation space: Arm+Head depth camera with relative position between object starting position and end-effector.
- Place $(s^*_i)$ Place the currently held object at goal state $s^*_i$:
  - Starting state: An object goal position and clutter is randomly spawned on one of 7 receptacles (sofa, fridge, counter left, counter right, light wood table, dark wood table, sink). The robot is facing the object goal with default noise applied to the base, orientation, and end-effector. The object to place starts in the robot's grasp.
  - Failure: $C_{max} = 7500$.
  - Success: The episode is a success if the object is at the goal and the arm is at the resting position.
  - Reward: $\mathbb{I}_{wrong}$ indicates the policy picked up an object.

$$r_t = 20\mathbb{I}_{success} + 5\mathbb{I}_{place} + 20\Delta^g_o\mathbb{I}_{holding} + 20\Delta^r_{arm}\mathbb{I}_{!holding} - \max(0.001C_t, 1.0)$$
$$-10\mathbb{I}_{force} - 5\mathbb{I}_{wrong}$$

  - Agent action space: Arm control. Once an object is placed, the gripper scalar action is masked out until the skill terminates.
  - Agent observation space: Arm+Head depth camera with relative position between object goal position and end-effector.
- Open fridge door $(s_i)$ open the door of the fridge containing object or goal position $s_i$:
  - Starting state: The fridge door starts closed. The robot spawns in a $0.9m \times 1.2m$ square in front of the fridge, facing the fridge handle with default noise applied to the base, orientation, and end-effector.

– Reward: $\mathbb{I}_{out}$ indicates the robot base left the spawn region.

$$r_t = 10\mathbb{I}_{success} + 5\mathbb{I}_{grabbed} + 1\Delta_{arm}^h + 1\Delta_a^g - 10\mathbb{I}_{out}$$

– Failure: There is no collision force threshold. The episode terminates with failure if the robot leaves the spawn region.
– Success: The episode is a success if the fridge is open more than 90 degrees and the robot is in the resting position.
– Agent action space: Arm and base control.
– Agent observation space: Arm+Head depth camera with relative position between end-effector and a target object starting or goal position in the fridge.

• Close fridge door $(s_i)$ close the door of the fridge containing object or goal position $s_i$:

– Starting state: The fridge door starts open with a fridge door angle in $[\pi/4 - 2\pi/3]$ radians. The robot spawns in a $0.9m \times 1.2m$ square in front of the fridge, facing the fridge handle with default noise applied to the base, orientation, and end-effector.
– Reward:

$$r_t = 10\mathbb{I}_{success} + 1\Delta_{arm}^h + 1\Delta_a^g$$

– Failure: There is no collision force threshold. The episode terminates with failure if the robot leaves the spawn region.
– Success: The episode is a success if the fridge is closed with angle within 0.15 radians of closed. and the robot is in the resting position.
– Agent action space: Arm and continuous base control.
– Agent observation space: Arm+Head depth camera with relative position between end-effector and a target object starting or goal position in the fridge.

• Open drawer $(s_i)$ open the drawer containing object or goal position $s_i$:

– Starting state: The drawer starts completely closed. A random subset of the other drawers are selected and opened between 0-100%. The robot spawns in a $0.15m \times 0.75m$ rectangle in front of the drawer to be opened, facing the drawer handle with default noise applied to the base, orientation, and end-effector.
– Reward:

$$r_t = 10\mathbb{I}_{success} + 5\mathbb{I}_{grabbed} + 1\Delta_{arm}^h + 1\Delta_a^g$$

– Failure: There is no collision force threshold.
– Success: The episode is a success if the drawer is between 90-100% open and the arm is at the resting position.
– Agent action space: Arm control.
– Agent observation space: Arm+Head depth camera with relative position between end-effector and a target object starting or goal position in the drawer.

• Close drawer $(s_i)$ close the drawer containing object or goal position $s_i$:

– Starting state: The target drawer starts between 80-100% open. A random subset of the other drawers are selected and opened between 0-100%. The robot spawns in a $0.15m \times 0.75m$ rectangle in front of the drawer to be closed, facing the drawer handle with default noise applied to the base, orientation, and end-effector.
– Reward:

$$r_t = 10\mathbb{I}_{success} + 1\Delta_{arm}^h + 1\Delta_a^g$$

– Failure: There is no collision force threshold.
– Success: The episode is a success if the fridge is closed and the arm is at the resting position.
– Agent action space: Arm control.
– Agent observation space: Arm+Head depth camera with relative position between end-effector and a target object starting or goal position in the drawer.

- `Navigate`: Navigates to the start of other skills. Importantly, the agent is only provided the 3D coordinate of the start or goal location to navigate to, for instance an object in the fridge or a location to place an object on the counter. This is different from the goal position the agent actually needs to navigate to which is on the floor in front of the object. The target on the floor is calculated based on the start state distribution of other skills. The agent does not have access to this privaledged information about the navigation goal position. Furthermore, the agent not only needs to navigate to a particular location but also face the correct direction (notated as $\theta^*$).

  - Starting State: A random base position and rotation in the scene. The state of the fridge, drawers, and object configurations are randomly sampled from one of the previous 6 skill training setups.
  - Reward:

$$r_t = 10\mathbb{I}_{success} + 20\Delta_{agent}^{goal} + \Delta_\theta^{\theta^*} I_{\Delta_{agent}^{goal}<0.9}$$

  Where $\Delta_{agent}^{goal}$ is the change in geodesic distance to the goal, $\theta$ is the current agent rotation, $\theta^*$ is the target orientation, and $\Delta_\theta^{\theta^*}$ is the change in L1 norm between the current agent angle and the target angle.

  - Failure: There is no collision force threshold. The episode horizon is $500$ steps.
  - Success: The agent is within 0.3 meters of the goal, 0.5 radians of the target angle, and has called the stop action at the current time step.
  - Agent action space: Similarily to [101], the navigation is handeled by a discrete action space which is then translated into continuous actions. Specifically, the linear velocity from -0.5 to 1 is discretized into 4 cells and the angular velocity from -1 to 1 is discretized into 5 cells, giving 20 cells in total. The action corresponding to 0 linear and angular velocity is the stop action.
  - Agent observation space: The Head depth camera with the relative position between the robot end-effector and object.

  We find that learning the termination condition is difficult for the navigation skill as demonstrated by Fig. 27 which demonstrates that learned termination results in a 20% drop in success rate.

### H.3 MonolithicRL

The **MonolithicRL** approach for the main task follows a similar setup as Appendix C but with a different action space and reward structure. The agent maps the egocentric visual observations, task-specification, and proprioceptive state into an action which controls the arm, gripper, and base velocity (policy architecture visualized in Fig. 24). The arm actions are the same as described in Section 5.

A challenge of the **MonolithicRL** approach is learning a long complicated task structure. We therefore train with a dense reward guiding the robot to complete each part of the task. Using a pre-specified ordering of the skills from Appendix H.2, we infer which skill the robot is currently at. We start with the first skill in the pre-specified skill ordering, when that skill terminates we progress to the next skill, etc. This current inferred skill only provides the reward to the **MonolithicRL** approach. The termination, starting state distribution, and transition function all still come from the actual task. We utilize a training set of 5000 configurations for the task. The evaluation set of task configurations consist of new objects placements.

## I Home Assistant Benchmark Further Experiments

### I.1 SPA Failure Analysis

In this section we analyze the source of errors for the **SPA** approaches for the HAB results from Fig. 5. Specifically, we analyze which part of the sense-plan-act pipeline fails. We categorize the errors into three categories. The first category ('Target Plan') is errors finding a collision free joint configuration which reaches the goal to provide as a goal state for the motion planner. The second category ('Motion Plan') is errors with the motion plan phase timing out (both **TP+SPA** and **TP+SPA-Priv** use a 30 second timeout). The third category ('Execution') is if the planned sequence of joint angles is unable to be executed. Failures for motion planning the pick, place and arm resets are grouped into these categories. These categories do not account for the learned navigation failure rates.

We analyze these sources of errors for **TP+SPA** and **TP+SPA-Priv** with learned navigation in Fig. 25. 'Target Plan' fails due to the sampling based algorithm timing out to find the collision free target joint state which accomplishes the goal. Methods therefore have a higher 'Target Plan' failure rate on `PrepareGroceries` where the agent must reach into the fridge to grab and place objects. **TP+SPA-Priv** has a higher 'Target Plan' failure rate because it has complete information about the geometry in the scene. This results in more obstacles being included in the collision check and therefore makes the target sampling harder. On the other hand, obstacles do not exist outside the perception of **TP+SPA** such as behind other objects or outside the field of view making the target sampling easier. Next, we see that all methods have a zero 'Motion Plan' failure rate. This means that when the algorithm is able to find a valid target joint state, the motion planning algorithm is able to find a valid series of joint configurations from the current joint state to the target joint state. Finally, the 'Execution' failure rates for **TP+SPA-Priv** is zero since this method uses a perfect controller. On the other hand, **TP+SPA** can fail to execute due to the imperfect execution controller and planning from incomplete information. A planned path returned as successful from the motion planner can fail due to unperceived obstacles.

### I.2   Learning Curves

All methods except for **MonolithicRL** utilize a set of skills. For **TP+SRL** these skills are learned with RL described in Appendix H.2. The learning curves showing the success rate as a function of the number of samples is illustrated in Figure 28. We include both the success rates from training and the results on a held out set of 100 evaluation episodes. **SPA** approaches use the robotics pipeline described in Appendix D and do not require any learning.

Since we found the Navigation skill difficult to train, we separately show the learning curves for the Navigation skill in Fig. 27. There we highlight the difficulty of learning the termination action by comparing to with and without the learned termination condition.

Likewise, we show the learning curves for the **MonolithicRL** approaches in Fig. 26. The success rate for picking the first object in `SetTable` is higher than `TidyHouse` since the object always starts in the same drawer for `SetTable`. Likewise, `SetTable` requires picking objects from an open drawer whereas `PrepareGroceries` requires picking objects from a tight fridge space.

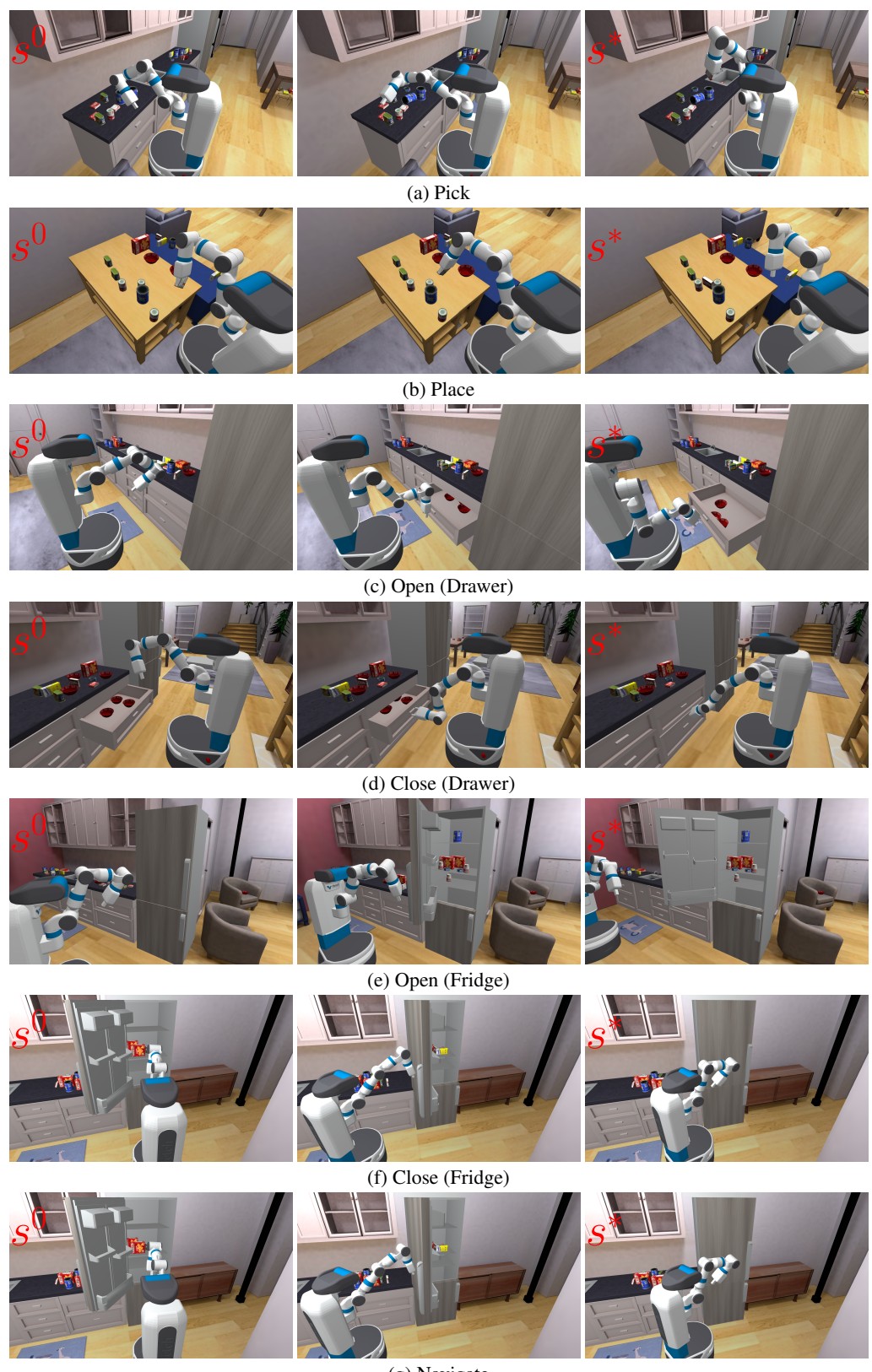

Figure 23: Overview of all the high level planner actions with the pre-conditions (right), post-conditions (left), and an intermediate state when executing the action.

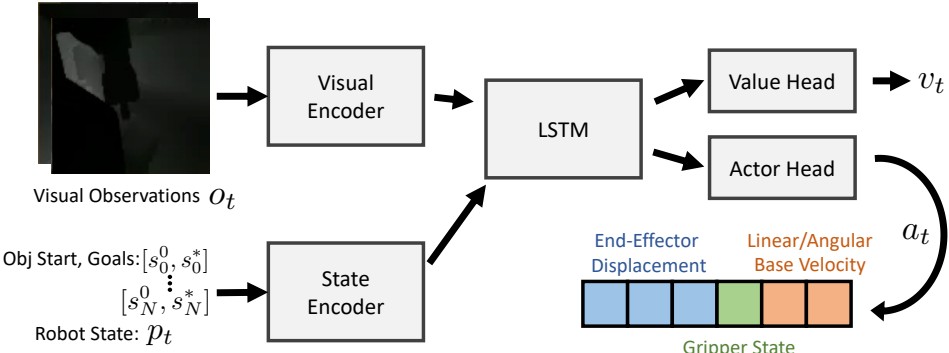

Figure 24: The **MonolithicRL** policy architecture for the HAB task. The policy maps egocentric visual observations $o_t$, the task-specification in the form of a series of geometric object goals $[b^1, g^1, \ldots, b^N, g^N$ where $N$ is the number of objects to rearrange, and the robot proprioceptive state $s_t$ into an action which controls the arm, gripper, and base velocity. A value output is also learned for the PPO update.

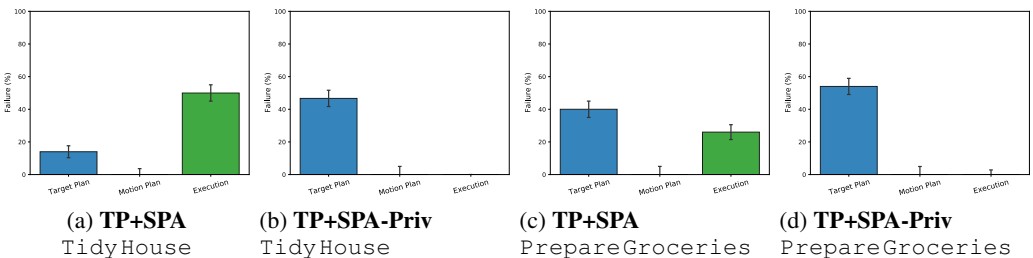

(a) **TP+SPA**
TidyHouse

(b) **TP+SPA-Priv**
TidyHouse

(c) **TP+SPA**
PrepareGroceries

(d) **TP+SPA-Priv**
PrepareGroceries

Figure 25: Motion planner failure rates for Fig. 5. Numbers indicate the percent of the 100 evaluation episodes the failure category occurs. 'Target Plan' is failures in finding a valid target joint configuration, 'Motion Plan' is the motion planning timing out, and 'Execution' is the planned sequence of joint angles failing to execute.

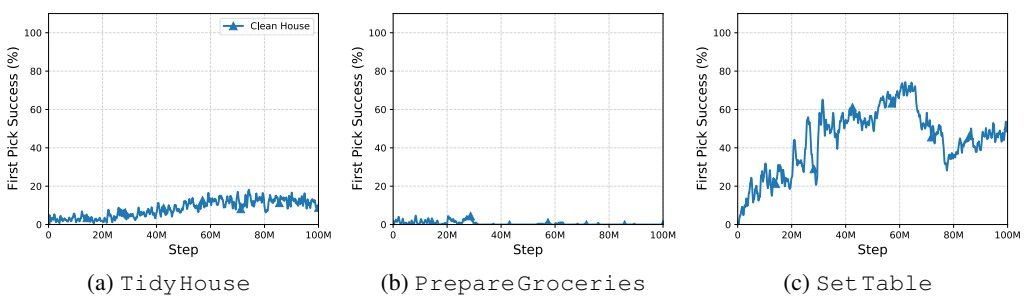

(a) TidyHouse

(b) PrepareGroceries

(c) SetTable

Figure 26: Training curves for the **MonolithicRL** approach for all tasks for a single seed. Y-axis shows success rates on picking the first object, in the case of TidyHouse this requires navigating to and picking an object from an unobstructed random receptacle, for PrepareGroceries this is navigating to and picking an object from the fridge, and for SetTable this is navigating to the drawer, opening it and then picking the object inside.

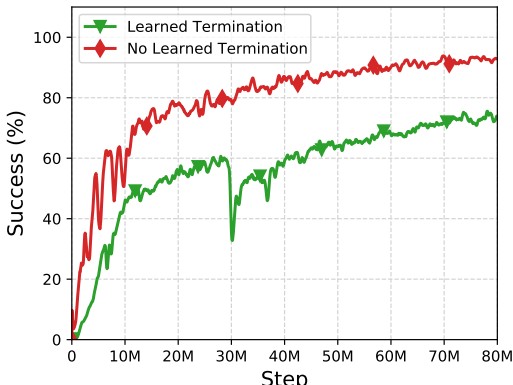

Figure 27: Training learning curve for the Navigation skill with and without the learned termination skill for 1 seed.

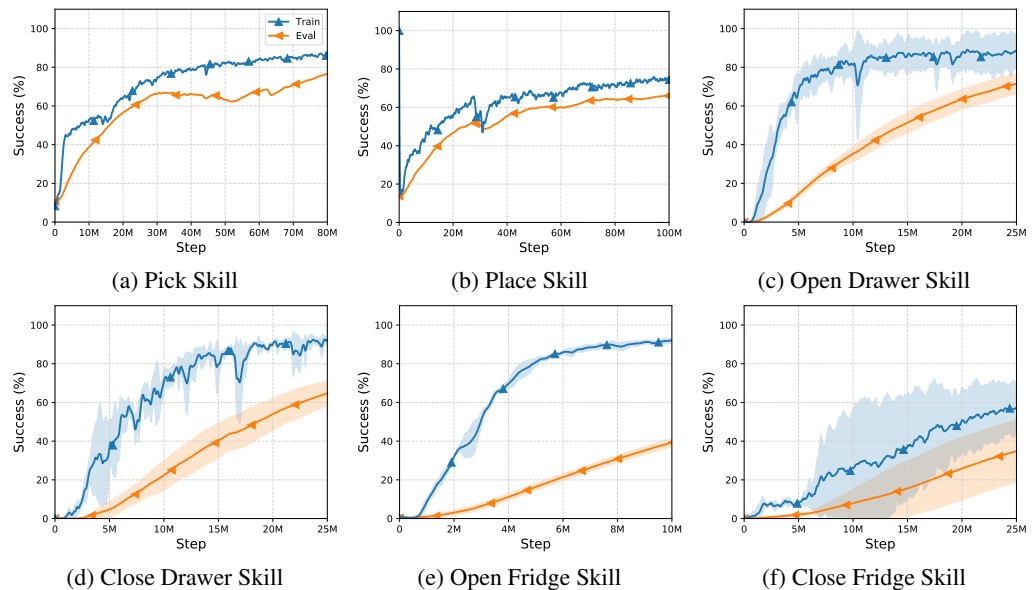

(a) Pick Skill

(b) Place Skill

(c) Open Drawer Skill

(d) Close Drawer Skill

(e) Open Fridge Skill

(f) Close Fridge Skill

Figure 28: Training and evaluation curves for the skills with averages and standard deviations across 3 seeds (except for the Pick and Place skills which are only for 1 seed).