# OpenReview forum: "Habitat 2.0: Training Home Assistants to Rearrange their Habitat"
_NeurIPS.cc/2021/Conference — NeurIPS 2021 Spotlight_

### Official Review · Reviewer_Pdk8 · 2021-07-17

**Rating:** 6
**Confidence:** 4

**Summary:**

The paper introduces a high performance simulation platform called Habitat 2.0 (H2.0) for training virtual robots and benchmarks RL and classical robotics policies on home assistant benchmark (HAB) that consists of three long-range tasks:  `tidying house` (moving 5 objects from random reachable locations back to where they belong), `Preparing Groceries` which is removing 2 objects from the fridge to the counters and place one back in the fridge and `setting table` which is about getting a bowl from the drawer, fruit from the fridge and placing fruit inside the bowl on the table.

The platform also involves a lot of engineering infrastructure development to increase the performance of the simulator leading to an impressive 850x improvement compared to real-time. It is worth noting that the grasping is abstracted i.e. whenever the gripper is within the 15cm range from the object, the object snaps into the gripper.

Using this simulator platform hey benchmark RL and classical robotics pipeline which they call as sense-plan-act (SPA) and show that SPA based policies are brittle but RL based policies need hierarchical planners as pure image to action based RL policies fail to solve long-range tasks due to various challenges involving the length of the task and inability to come up with a sensible reward function. However, RL with hierarchical planners aren't perfect either and can suffer from inability to cope with the complexity of the environment. Therefore, the simulation platform provides a humbling peek into the inability of current state of the art algorithms.

**Limitations And Societal Impact:**

While exploring the robotics set-up, the paper seems to be mostly focussing on the understanding of the RL algorithms and cares less about the real problems involved in doing learning for robotics e.g. grasping is abstracted with considerable and generous tolerances of 15cm and that if the gripper is within this tolerance the objects snaps into it. Most often in the real world, the issues arise due to unstable or impossible grasping and that is abstracted completely in this work. In the end even if we have perfect planner designing generalisble low-level controllers is pretty hard. This work seems to completely sidestep that part. Obviously this makes the problem even more challenging but I'd have liked to see some work in that direction too. The success rate would go down quite a bit but it would be worth plotting. Moreover, as more contacts are made, the simulator will slow down too. Table 1 shows comparison against various other simulators but they don't all necessarily have similar settings so it is not fair.

Also, when they simulate things at 30Hz (time step 1/30) there is no motion blur simulation which could also affect the policy. I wonder if there was any decision choices made for that I didn't see that in the paper. In the real world motion blur is certainly something that is going to affect the performance of the algorithm.

**Main Review:**

- The paper shows that pure image to action policies --- called Monolithic RL --- are simply not capable to solving tasks on their own. This is because of the complexity of the task that is well beyond the reach of current SOTA RL algorithms as well as the reward function. In general, for long-range tasks, coming up with the right objective remains elusive.

- Comparison between TP+SRL --- which is policies involving TaskPlanning + SkillsRL where TaskPlanner (STRIPS) is used to break the task down into atomic tasks which are all learned via RL --- and TP+SPA --- (STRIPS planner) and sense plan act policy involving classical robotics pipeline to solve the tasks --- doesn't reveal a straightforward pattern. On the `TidyHouse` task, TP+ SPA performs better than TP+ SRL while the trend is reversed for the `PrepareGroceries` task.

It is humbling to see that even for seemingly simpler tasks with near-perfect planners guiding the chain of skills, it's not possible to fully solve the tasks requiring multiple interactions with the environments. This is even more surprising considering that the grasping has been abstracted in this work. The issue seems to be either compounding errors or agents reaching positions where the are unable to view the required workspace in their viewport. Providing an oracle navigation helps understand the gap as they tend to have better success rates.

The engineering effort in building up the pipeline and platform to carry out tasks at extreme high frame-rates is very commendable.

**Time Spent Reviewing:**

2

---

> ### Author Response · Authors · 2021-08-10
> **Response to Reviewer Pdk8**
>
> **1. Comparison between TaskPlanning+SkillsRL (TP+SRL) and TaskPlanning+SensePlanAct (TP+SPA) doesn't reveal a straightforward pattern.**
>
> We respectfully disagree and believe there is a trend. As we describe in L388-395, Figure 5 shows the pattern that TP+SRL performs better relative to TP+SPA as the task complexity increases (L388-395). In the easiest setting (Tidy House with oracle navigation) TP+SPA performs better than TP+SRL, however, as task difficulty increases (Tidy House no oracle navigation) TP+SRL performs better and in an even harder setting (Prepare Groceries) TP+SRL performs much better.
>
> **2. Show success rates with full, non-abstracted grasping.**
>
> Happy to. We removed the abstracted grasping and ran experiments for the Pick task with full grasping and found that policies are still able to learn meaningful Picking behaviors. For full grasping, the policy outputs a single continuous value which sets the separation of the parallel gripper. We use the same experimental setup described in Appendix E.2, with two seeds and a policy trained over 100M steps of experience. We find the success rate without abstracted grasping is 20.16% versus 89.70% with abstracted grasping. This large gap is expected since the policy must grasp from 9 randomly sampled and randomly placed YCB objects. Learning such diverse grasps is a challenging problem in of itself and an active area of research [1,2]. We will include this result in the supplementary. Thank you for the suggestion.
>
> **3. Can we include motion blur simulation?**
>
> Yes, happy to. We agree that motion blur is an important factor that can affect real-world performance. We therefore added motion blur simulation using the Kornia library and computed the blur parameters based on the camera movement [4]. We trained a pick policy on random motion blurs and found that the policy learns to pick just as successfully without the motion blur (88.42% vs 89.83% success). We also tried training navigation policies with motion blur but found learning to be unsuccessful. Unlike for picking, navigation must learn the termination condition, which is difficult as highlighted in Appendix L1268-1270+Figure 28. The motion blur makes learning this task even harder and is consistent with findings by recent work [3].
>
> Learning policies that are robust to visual perturbations such as motion blur is an interesting direction. However, a deeper investigation is outside the scope of our paper.
>
> **Works Cited**
>
> [1] Mousavian et al. "6-dof graspnet: Variational grasp generation for object manipulation." ICCV 2019.
>
> [2] Murali et al. "6-dof grasping for target-driven object manipulation in clutter." ICRA 2020.
>
> [3] Chattopadhyay et al. “RobustNav: Towards Benchmarking Robustness in Embodied Navigation.” arXiv:2106.04531 (2021)
>
> [4] Riba, Edgar, et al. "Kornia: an open source differentiable computer vision library for pytorch." Proceedings of the IEEE/CVF Winter Conference on Applications of Computer Vision. 2020.

---

> ### Author Response · Authors · 2021-08-20
> **Has Our Response Addressed Your Concerns?**
>
> Dear Reviewer Pdk8, we would be grateful if you can comment on whether our response addressed your concerns or if issues remain. To summarize our response:
> - We clarified that the pattern between TaskPlanning+SensePlanAct (TP+SPA) and TaskPlanning+SkillsRL (TP+SRL) is that TP+SRL performs better than TP+SPA as the task complexity increases.
> - We showed success rates with fully simulated, non-abstracted grasping.
> - We added motion blur and compared performance on the Pick and Navigation skills.

---

> > ### Comment · Reviewer_Pdk8 · 2021-08-24
> > **The rebuttal addressed my concerns**
> >
> > Yes, the rebuttal addressed my concerns. I'm happy that you did those experiments and it is nice to see that your results and conclusions still hold true for the motion blur experiments. Without abstracted grasping I think it's expected that your performance drop will be significant and it is worth emphasising that in the paper.
> >
> > Minor comment: I did not disagree with you in my review with regards to the trend you observed. I just felt the trend was reversed and it needed more explanation. I think it's important to have characterisations and readers can appreciate more what you have done.
> >
> > Overall, I think this is a work in the right direction and I fully support such work. Thank you.

---

### Official Review · Reviewer_Xe9G · 2021-07-19

**Rating:** 6
**Confidence:** 4

**Summary:**

This paper presents a new simulation environment for robotic pick-and-place tasks. The main contribution is a significant engineering effort to accelerate simulation around an existing physics engine, Bullet. The work includes some models for simulation and three tasks to test solutions. The paper also evaluates “flat” reinforcement learning, and hierarchical structure, with learned subcomponents or motion planning ones.

**Ethical Concerns:**

Beyond the continuously listed “hours/months of work”, which I consider not a great element to highlight from a company, I do not see other ethical concerns on this work.

**Limitations And Societal Impact:**

I see some limitations and unclear parts in the current manuscript that I list below. I hope this helps to improve the manuscript.

It is a bit unclear what is the number of apartments. “111 unique layouts of a single apartment background”. I’m interpreting this as “one apartment, 111 different ways of arranging the furniture”? In that case, it sounds limited with respect to AI2Thor, RoboThor (that enables multiple arrangements) and iGibson (that provides 15 environments and the possibility to change the objects). TDW includes 100 rooms. From a user’s perspective, why not unifying all of those? I get that AI2Thor is a bit harder because it is Unity, but iGibson is also Bullet? 92 objects do not sound very large as dataset. AI2Thor includes more than 100 objects. iGibson annotated 150 models. While valuable, I think to be a strong contribution, a dataset of objects should include many more objects, e.g., ShapeNet, PartNet…

It is a bit strange to continuously highlight the time invested in the different parts (hours of work, years of work). I don’t think this should be a factor to decide on the value of something. I would prefer to remove references to that, especially assuming that a large company is behind this effort, and making the “hours of work” a feature to highlight would clearly benefit large corporations with large teams or the budget to hire contractors, which could be an ethical concern.

When explaining Habitat 2.0, the text explains the capabilities of any physics simulator, and that makes it hard to attribute credit. First, I think it should be necessary to cite the first time (line 35) the true source of the code, which is Couman’s Bullet because H2.0 is not advancing the field of rigid body simulation, the core equations are the same. Second, highlighting rigid body simulation capabilities in the way they are listed may be misleading: any rigid body physics simulator can do all of this, and if it supports simulating kinematic chains it can simulate any robot. Are all the listed robots modeled and included in H2 or are they just random examples? I couldn't find in the code. Please, clarify.

The number of tasks is somewhat also not to pair with other benchmarks like metaworld, or RLBench. The way of defining the tasks is also very ad-hoc: there is a new term used “GeometricGoal”, but this is the common way of defining pick and place tasks in robotics: place the object at this location with some threshold. How scalable is the definition of tasks? Does it have to be manually defined for each new task? Can the tasks be defined more semantically instead of geometrically?

The related work section is lacking depth. Currently, there is a lot of simulation environments out there and it is increasingly difficult for users to understand their pros and cons, so I really think this section is critical. For example, I would recommend analyzing in detail the “closest neighbors”: AI2Thor (and variants), Sapien, iGibson, ThreeDWorld, RLBench. Right now, in two paragraphs, all physics simulators, renderers, and game engines are mentioned, but this feels more like a list than an in-depth comparison. The paper mentions several “points” that this work contributes but the related work focuses only on one: speed. In that sense, it feels like this is really the only difference to existing work. The subsections of the related work are also strange as they try to explain something instead of placing the current work in a scholarly context. The comparing table is a great idea but the information included right now is not very useful: it includes the names of the rendering libraries and the type of rendering (not explained or referred in the paper) but the only critical number is the speed, lacking a comparison in the other mentioned contributions (the scenes/objects, and the benchmark tasks). Also, the cited paper [33] is really similar, it would be great to go deeper in the comparison. Although it is only arxiv, I would recommend including a comparison to “The ThreeDWorld Transport Challenge” that is also extremely similar.

For related work, would be possible to include name+number instead of only the number? E.g. “Garret et al. [55] study…” instead of “[55] study…”. This facilitates readability.

Joint type and limits are not dynamic properties, they are kinematics.

It is a bit unclear to me why highlighting and novel the “Localized Physics”. Bullet (as most mature physics simulators) already optimize in this front. Bullet uses the well-known concept of “islands” of objects to simulate and activate/sleep bodies. This is a default functionality. The only thing I can infer from this text is that this project may have improved a bit on that? Still, the way it is explained (“ if a robot is picking a mug from a living-room table, why must we check for collisions between the kitchen fridge shelf and objects on it?”) is to be misleading as it sounds like a novel contribution while it is only a (valuable) engineering fine-tuning of an existing concept and implementation. The second contribution listed there, “using a navigation mesh to move the robot base kinematically”: how does this affect if there are objects on the floor, for example, one of the picked objects is dropped? Can the objects be dropped? Maybe the “perfect grasping” impedes that? It would be interesting to deactivate the “perfect grasping” functionality in a small experiment (maybe the pick one) and evaluate the performance. This would tell us how realistic the results are.

Why is there an association between the “simulation step” and the “control step”? Is the control signal changed at 120Hz? Is there a controller running? If not, it is just physics simulation, right? Please, clarify.

It would be great to have a discussion on parallelization. What is parallelized? The same environment? How is that possible? Copies of the same environment? I wonder how the use of parallelizing computers and GPUs relate to other simulators, for example, to Mujoco, concretely with respect to their analysis here: http://www.mujoco.org/performance.html and in related papers.

Does the robot have access to the absolute location of the base? This is given as “realistic proprioception” but I’m not sure this is the default on any robot. GPS won’t provide this level of accuracy, even if it would work in an apartment.

Why measuring the accumulated contact force in the experiments and use it to stop? What is that physically? Is the accumulation over the entire arm at a time step (in that case, 5000N seems way off the payload) or over time? If it is over time, there is no limit per time step?

I’m confused about the Sense-Plan-Act paradigm listed as “classical”. The alternative tested does task planning and uses RL-learned skills but is also doing the same three “primitives”. I went back to Murphy’s book and I do not understand the terminology in the paper. Maybe the RL option can be also explained in the same terms as the cited Murphy book? Is the main difference not just that the “act” part is executed by a learned policy? For example, in line 286, the solution explained is a “Sense-Act” (Murphy calls this “reactive paradigm”).

How are collisions implemented? Are there fully implemented? Some visualizations of the trajectories show the arm going through objects. Also, how what is the field of view of the cameras? Are they realistic?


**Main Review:**

Originality: The presented work presents a version of the simulator and novel models. However, intellectually, I do not appreciate considerably novel contributions. Different forms of the features presented here have been presented by others before (models of scenes and objects, interactive simulation, rearrangement tasks,…). In that sense, the paper does not present original ideas or simulation concepts.

Quality: The text, images, and code are well executed. I appreciate the work of a large engineering group, as it is highlighted explicitly several times (see my comment on repeating the invested time in the “limitations” section).

Clarity: The paper is clearly written. However, there are some concepts that are presented in a way that sounds like novel contributions of this work instead of part of the codebase this work extends (like Bullet functionalities for sleeping objects, or the common rigid body physics simulation). It would be great to 1) make clear in the text what is really novel in this work, 2) compare to state of the art in those aspects. Any element that is listed as a contribution, should be compared in the related work.

Significance: The code presented here (the new simulation environment) could be useful for the community. The manuscript does contain some additional information.

In summary, I see the engineering value of the work, but the intellectual novelty is not high.



**Time Spent Reviewing:**

6

---

> ### Author Response · Authors · 2021-08-10
> **Response to Reviewer Xe9G**
>
> **1. The main contribution is a significant engineering effort to accelerate physics simulation by modifying the code base of a well-known simulator, Bullet, to be better parallelized and optimized.**
>
> This claim is factually incorrect.
>
> First, the reviewer is ignoring our other contributions of the ReplicaCAD dataset (Section 3) and all our research findings on the Home Assistant Benchmark tasks (Sections 5,6). All other reviewers explicitly acknowledge these other contributions.
>
> Second, even if we focus on the Habitat 2.0 simulator (Section 4), as described in L180-183, our contributions are -- localized physics and rendering (Appendix B.1), interleaved physics and rendering (Sec. 4.1), simplify-and-reuse (Appendix B.2), and integration with motion planning libraries (Appendix B).
>
> Nearly all of these contributions live outside the code-base and scope of Bullet, so a claim that “the main contribution … [is] modifying the code base of [Bullet]” is simply wrong. It is like saying that a typical ML paper simply modifies the code-base of PyTorch or TensorFlow. Most of the work happens outside the tool, not within it.
>
> **2. The features presented (models of scenes and objects, interactive, simulation, rearrangement, solutions,…) have been presented before in several forms.**
>
> This claim is factually incorrect.
>
> Our datasets (models of scenes and objects), simulator, benchmark tasks, and experimental analysis have NOT been presented elsewhere. Making a claim about lack of originality without citing where and how something has already been presented is poor reviewing practice.
>
> **3. H2.0 is not advancing the field of rigid body simulation, the core equations are the same.**
>
> We do NOT claim to advance the state of art in rigid-body mechanics.
>
> We do claim that Habitat 2.0 simulator makes several optimizations which result in a 100x simulation speedups over prior work. We achieve this not through modifying Bullet, which we never claim in the paper, but instead through optimizing the use of Bullet specifically for embodied AI house tasks, a tight coupling of rendering and physics, and an optimized asset pipeline (details in Section 4). Furthermore, Habitat 2.0 is just one of our three major contributions with the other two being ReplicaCAD (Section 3) and the Home Assistant Benchmark and associated analysis (Section 5,6).
>
> **4. Why highlight “Localized Physics” and claim this as novel? Bullet uses the well-known concept of “islands” of objects to simulate and activate/sleep bodies. This is a default functionality.**
>
> We do NOT claim Bullet’s island (or any other existing) functionality as our own.
>
> As described in Appendix B.1, our contributions to Localized Physics are in all cases either unrelated to this system or complementary to it.
>
> In particular, we made a significant change to how multi-body articulated furniture like kitchen cabinets are loaded into Bullet; this allows the existing island/sleep system to work better for these objects. We also implemented an optimization that leverages the rigid-body sleep state to speed up our custom renderer. Finally, our navigation-mesh abstraction is entirely unrelated to the island/sleep system.
>
> **5. Experiment with deactivating the “perfect grasping” and use full grasping.**
>
> This is a good question. We removed the “perfect grasping” and ran experiments for the Pick task with “full grasping” and found that policies are still able to learn meaningful picking behaviors. With full grasping, the policy outputs a single continuous value which sets the separation of the parallel gripper. We use the same experimental setup described in Appendix E.2, with two seeds and a policy trained over 100M steps of experience. We find the success rate with full grasping is 20.16% versus 89.70% with perfect grasping. This large gap is expected since the policy must grasp from 9 randomly sampled and randomly placed YCB objects. Learning such diverse grasps is a challenging problem in of itself and an active area of research [7,8]. We will include this result in the supplementary. Thank you for the suggestion.
>
> **6. The way of defining the tasks is also very ad-hoc: there is a new term used “GeometricGoal”, but this is the common way of defining pick and place tasks in robotics: place the object at this location with some threshold.**
>
> We agree with the reviewer that we follow standard robotic practices in defining a task through start and desired goal position for all objects. As we describe in L59, L75, L122, L281 we follow the naming convention and recommendations of a recent position and survey paper released by a large multi-institution multi-disciplinary group of researchers.
>
> **7. How scalable is the definition of tasks? Does it have to be manually defined for each new task? Can the tasks be defined more semantically instead of geometrically?**
>
> As we describe in L164-L176, our tasks can be procedurally generated and thus are scalable. Our experiments are evidence for this with unique start and goal states for 5k training episodes and 100 evaluation episodes. Tasks are defined semantically through the ReplicaCAD receptacle annotations (L164) which enable sampling positions based on a receptacle name. GeometricGoal problems are instantiated from a semantic definition of the object starting receptacle and the object goal receptacle.
>
> **8. Comparison with “The ThreeDWorld Transport Challenge”**
>
> We already compare to TDW in Table 1 (upon which the TDW Transport Challenge is based). As we elaborate in L245, TDW achieves 5-168 steps per second (SPS) which is orders of magnitude less than the 1400 SPS in Habitat 2.0.
>
> Diving deeper into the TDW Transport Challenge, our work and TDW support the same conclusion that end-to-end RL baselines and hierarchical baselines struggle on rearrangement tasks in the home. To control the robot, our experiments use a 10 dimension continuous action space whereas TDW uses 6 discrete actions. We also train policies for an order of magnitude more experience (100M samples vs 10M samples) and we compare with sense-plan-act architectures which use motion planning. To their credit, TDW includes containers which the agent can learn to use as a tool to more efficiently transport multiple objects. The emergence of tool use is an exciting future avenue to explore in Habitat 2.0.
>
> **9. Also, the cited paper [33] is really similar, it would be great to go deeper in the comparison**
>
> We already do this in L134-L139. The ManipulaTHOR framework from [33] does not involve interactions with container objects and only requires rearranging a single object. ManipulaTHOR is also listed in Table 1 with speeds of 30-60 steps per second (SPS) compared to the 1400 SPS in Habitat 2.0.
>
> **10. Why is there an association between the “simulation step” and the “control step”? Is the control signal changed at 120Hz?**
>
> A control step is 4 physics simulation steps + 1 render call (line 228). No, the control signal is changed at 30Hz and simulation steps at 120Hz (line 228).
>
> **11. What is parallelized? I wonder how the use of parallelizing computers and GPUs relate to other simulator**
>
> We already describe our parallelization setup in L246-L249 and Appendix B.3. For parallelized benchmarking, we run multiple (independent) processes, where each process has one instance of H2.0 simulator. This benchmarking is designed to show that a single instance of H2.0 is both fast and lightweight, allowing us to extract more performance from the overall system via parallelization.  This style of parallelization is consistent with the other simulators discussed such as iGibson, AI2Thor, and TDW.
>
> **12. Does the robot have access to the absolute location of the base? This is given as “realistic proprioception” but I’m not sure this is the default on any robot.**
>
> No. As described in L268, we use a base egomotion sensor. The robot does not have access to the absolute location of the base. This is a common assumption in navigation and similar simulators [2,9,10] since egomotion by itself is a challenging problem [3].
>
> **13. Why measure accumulated contact force and what does this mean?**
>
> The accumulated contact force is to help mitigate damage to the robot and surrounding scene. As described in L283-L285 and Appendix C.2 the accumulation is over the entire robot and scene during an episode. There is no limit per time step. We use the accumulated force as a heuristic for damage to the robot and scene. For instance, repeatedly hitting the arm against the fridge even with low impact force, can cause arm damage. Other termination conditions, such as a maximum instantaneous force as suggested by the reviewer, could be added to help improve the safety of the robot.
>
> **Works Cited**
>
> [1] Batra et al. "Rearrangement: A challenge for embodied ai." arXiv:2011.01975 (2020).
>
> [2] Savva, et al. "Habitat: A platform for embodied ai research." ICCV 2019.
>
> [3] Datta et al. "Integrating Egocentric Localization for More Realistic Point-Goal Navigation Agents." arXiv:2009.03231 (2020).
>
> [4] Deng et al. "Imagenet: A large-scale hierarchical image database." CVPR 2009.
>
> [5] Gan et al. "Threedworld: A platform for interactive multi-modal physical simulation." arXiv:2007.04954 (2020).
>
> [6] Gan et al. "The ThreeDWorld Transport Challenge: A Visually Guided Task-and-Motion Planning Benchmark for Physically Realistic Embodied AI." arXiv:2103.14025 (2021).
>
> [7] Mousavian et al. "6-dof graspnet: Variational grasp generation for object manipulation." ICCV 2019.
>
> [8] Murali et al. "6-dof grasping for target-driven object manipulation in clutter." ICRA 2020.
>
> [9] Chaplot et al. “Learning To Explore Using Active Neural SLAM” ICLR 2020.
>
> [10] Chaplot et al. “Object Goal Navigation using Goal-Oriented Semantic Exploration.” NeurIPS 2020.

---

> > ### Comment · Reviewer_Xe9G · 2021-08-31
> > **On the replies**
> >
> > Thank you for the clarifications and the additional experiments. I will modify my review based on those.
> >
> > Re1:
> > It is true that the “by modifying the code-base” may not be completely accurate. I will change it to  “The main contribution is a significant engineering effort to accelerate simulation around an existing physics engine, Bullet.”.
> >
> > With respect to the main part of my sentence, I still consider that the main contribution is the acceleration of the simulator. I don’t see how this can be “factually incorrect”. I’m not “ignoring the other contributions”, I just consider them to not be the main contribution. I never said it is the “only” contribution.
> >
> > With respect to the other contributions, the authors seem to be ignoring my comments about comparing with existing work to understand the value of the new features. I think this would improve the paper and make clear the value of the work. There are many datasets of scenes and objects (TDW, AI2thor, iGibson, Isaac, VirtualHome, Sapien/Partnet,...), and also many benchmarks of activities (Metaworld, RLBench, Transport challenge, the control ones like DeepMind control suite, OpenAI Gym, ...). There is no comparison to these ones, so the reader cannot sure what their limitations are that get covered by ReplicaCAD and the HAB. Comparing and understanding them would help to understand the value of the presented work. Table 1 is a good opportunity, but since the only feature compared there is the speed (and code libraries), this only supports the idea that the main contribution is to accelerate the simulator.
> >
> > Re2:
> > This comment was misinterpreted, I will make it clearer in the review. It is not that these particular models have been presented before, as the authors discuss in the rebuttal, but that the concepts are similar to previously presented ones in different sizes and forms, but not compared to those in the most relevant dimensions.
> > Models of scenes and objects: TDW [1], AI2thor [2], iGibson [3], Isaac [4], VirtualHome [5], Sapien/Partnet [6],...
> > Taking a 3D scan building a CAD version of it: Scan2CAD [7], iGibson [3], RoboThor [8].
> > Other rearrangement tasks: TDW challenge [9], VisualRoom rearrangement [10], Rearrangement arxiv paper [11].
> > I agree that it is hard to make very different contributions in this field (although some great recent ones do, e.g., DiSECt [12] or MegaVerse [13]), but I think it would be great for the field to make an honest effort comparing to existing and very similar alternatives. Increasing the Table 1 would be great, maybe removing the names of the libraries to gain space.
> > [1] Gan, Chuang, et al. "Threedworld: A platform for interactive multi-modal physical simulation." arXiv preprint (2020)
> > [2] Kolve, Eric, et al. "Ai2-thor: An interactive 3d environment for visual ai." arXiv preprint (2017).
> > [3] Shen, Bokui, et al. "iGibson, a Simulation Environment for Interactive Tasks in Large Realistic Scenes." arXiv preprint (2020).
> > [4] Makoviychuk, Viktor, et al. "Isaac Gym: High Performance GPU-Based Physics Simulation For Robot Learning." arXiv (2021).
> > [5] Puig, Xavier, et al. "Virtualhome: Simulating household activities via programs." CVPR. 2018.
> > [6] Xiang, Fanbo, et al. "Sapien: A simulated part-based interactive environment." CVPR. 2020.
> > [7] Avetisyan, et al. "Scan2cad: Learning cad model alignment in rgb-d scans." CVPR. 2019.
> > [8] Deitke, Matt, et al. "Robothor: An open simulation-to-real embodied ai platform." CVPR. 2020.
> > [9] Gan, Chuang, et al. "The ThreeDWorld Transport Challenge: A Visually Guided Task-and-Motion Planning Benchmark for Physically Realistic Embodied AI." arXiv preprint, (2021).
> > [10] Weihs, Luca, et al. "Visual Room Rearrangement." CVPR. 2021.
> > [11] Batra, Dhruv, et al. "Rearrangement: A challenge for embodied ai." arXiv preprint (2020).
> > [12] Heiden, Eric, et al. "DiSECt: A Differentiable Simulation Engine for Autonomous Robotic Cutting." RSS, 2021.
> > [13] Petrenko, Aleksei, et al. "Megaverse: Simulating Embodied Agents at One Million Experiences per Second." ICML, 2021.
> >
> > Re3:
> > The sentence is quoted out of context. The real context: “First, I think it should be necessary to cite the first time (line 35) the true source of the code, which is Couman’s Bullet. H2.0 is not advancing the field of rigid body simulation, the core equations are the same.” I will change it to not miss the context.
> > The sentence from the manuscript where I think Couman’s work should be cited is:
> > “H2.0 supports piecewise-rigid objects (e.g. door, cabinets, and drawers that can rotate about an axis or slide), articulated robots (e.g. mobile manipulators like Fetch [1], fixed-base arms like Franka [4], quadrupeds like AlienGo [5]), and rigid-body mechanics (kinematics and dynamics).”
> > Here (and in other points, see below) the manuscript is unclear on what are new contributions and what is inherited from others. I think it is a good scientific practice to work hard to make clear these blurry lines and give the right credit.
> > I previously asked if these robots mentioned are supported or just named randomly. I searched for it in the submitted code but couldn’t find them, only Fetch, but maybe I didn’t find the right files. Since making a robot available in a simulator is a significant effort beyond importing the URDF, it would be great to make clear what robots that are currently part of H2.0.
> >
> > Re4:
> > I never said the “authors claim the concept of islands as theirs”.
> > But, again, here the manuscript is unclear and makes it look as there is nothing in place to address that in Bullet. The confusing part: “Simulating physics for every part at all times is not only slow, it is simply unnecessary – if a robot is picking a mug from a living-room table, why must we check for collisions between the kitchen fridge shelf and objects on it?” That is exactly what islands achieve, to not have to simulate physics for every part at all times. However, the concept of islands is never mentioned. If there is an existing mechanism to achieve this but is not good enough, a good scientific practice is to explain the limitation of the existing mechanism to better understand the contribution of the new one. If it is complementary, how do they complement each other? What does one achieve that the other does not? Mentioning the existence of a mechanism in place to not “check for collisions between the kitchen fridge shelf and objects on it” that is not enough for some reasons (and explaining why), would help the readers understand why the new mechanism this necessary, and even consider it a good general alternative that other simulators could use to complement islands.
> >
> > Re5:
> > This is interesting. Thank you for the additional experiments, they are valuable! It seems to keep a similar trend between the trained and the motion-planning low-level policies.
> >
> > Re6:
> > OK
> >
> > Re7:
> > That’s interesting. Including this in the supplementary material would be valuable.
> >
> > Re8:
> > The TDW challenge is about picking objects and bringing them to locations. Beyond speed, it would be great to understand if there are conceptual differences. The included comparison is informative and I’m grateful for it, also about the comparison between solutions. I wonder how their action primitives using motion planners relate to the ones used here in the experimental evaluation. I’m not sure if TDW challenge includes closed containers for objects, maybe that is another difference.
> > It would be good to compare to TDW (and the other similar benchmarks) in the other dimensions where this manuscript mentions doing contributions: object and scene models, different tasks, or variations of them.
> >
> > Re9:
> > Is there any limitation in ManipulaThor to manipulate containers or is it just not shown in the experiments? I’m trying to understand if the limitation is in the simulator or in what they proposed as tasks.
> >
> > Re10:
> > The sentence creating the confusion is: “Each simulation step consists of 1 rendering pass and 4 physics-steps, each simulating 1 /120 sec for a total of 1 /30 sec. This is a fairly standard experimental configuration in robotics (with 30 FPS cameras and 120 Hz control).”. This seems to assume the physics step and controller step are the same. However, in some simulators, they are different, as the controller frequency is bounded by computation speed on the real robot and specs of the sensors used for the feedback loop, while simulation timestep can decrease as much as desired to obtain more accurate results.
> > By the way, the user guide of Bullet mentions: “Warning: in many cases it is best to leave the timeStep to default, which is 240Hz.” I wonder if you observed simulation artifacts from decreasing the simulation frequency to half the recommended? Are the other compared simulators using the same timestep used in H2.0 or the recommended one by the physics engine?
> > "No, the control signal is changed at 30Hz and simulation steps at 120Hz”. I assume this sentence is wrong. I guess it means that the goal for the controller changes at 30Hz but the controller computes with the same goal and new sensor signals (perfect joint encoders) to produce new control signals at 120Hz.
> >
> > Re11:
> > Thank you, that is clearer.
> >
> > Re12:
> > Is this perfect egomotion sensing, i.e., is there noise in the egomotion? I agree egomotion/odometry (e.g., visual) is a very hard problem; controlling the navigation commands based on perfect measurement of the egomotion makes things easier.
> >
> > Re13:
> > Manufacturers of real robots usually report the payload allowed, which is immediate maximum force (or weight) allowed. The proposed measurement would consider equally good/bad to apply 1000N in one step and 0 in the next 999 steps than applying 1N over 1000 steps. I find it valuable to have considerations over the applied force, and the accumulation over the entire robot provides some safety, I’m trying to understand why not just penalizing all large forces. Maybe is because the instantaneous forces from Bullet are not accurate enough?

---

> > > ### Author Response · Authors · 2021-09-02
> > > **Response to Reviewer Xe9G**
> > >
> > > We thank the reviewer for their detailed response and address their comments below:
> > >
> > > **Re1: “The main contribution is a significant engineering effort to accelerate simulation around an existing physics engine, Bullet”**
> > >
> > > This is indeed closer to the truth. However, please note that our 100x improvements are not just about physics; but involve a sophisticated interplay between rendering and physics (Sec 4.1) as supported by the impact of rendering optimizations in Table 2 which improve performance by over 50%, improving one of the benchmarks from 18k SPS to 25k SPS. Only optimizing physics as the reviewer states will simply shift the bottleneck to rendering due to the visual complexity of our environments, leaving the overall simulation speed slow.
> > >
> > > **Re1,2: ReplicaCAD vs Prior Datasets**
> > >
> > > First, several of the works mentioned by the reviewer (VirtualHome [10], IsaacGym [21]) do NOT publicly release their assets and thus there is nothing for us to compare to. Similarly, TDW [5] has not yet publicly released the majority of their assets, so we cannot test or comment on them.
> > >
> > > Second, PartNet assets from Sapien [9] were scraped from Google warehouse assets and it is unclear if the licensing terms associated with those assets allow for research use. As described in L32-34, ReplicaCAD was created in compliance with artist contracting laws and we have the legal rights to distribute the data under the permissive Creative Commons license. Similar to our work in spirit, iGibson features synthetic scenes based on real-world spaces, but it also uses the object models from PartNet inheriting these licensing issues.
> > >
> > > Next, while some objects from AI2-Thor and ManipulaTHOR [6] support opening and closing, these assets are built for a simulator that only changes object states through scripted animation, meaning the robot cannot open or close them via the manipulator. Finally, unlike our work, none of the assets introduced in these prior works have supporting surface annotations for procedural object clutter generation.
> > >
> > > **Re1,2,8,9: Home Assistant Benchmark (HAB) vs Prior Tasks**
> > >
> > > We will answer the reviewer’s question, but please note that DiSECt [11] appeared on ArXiV on May 25, which is after the NeurIPS paper registration deadline. Megaverse [12] appeared on ArXiV on July 17, 2 months after the NeurIPS submission deadline. And IsaacGym [21] was released on Aug 24, 3 months after the NeurIPS submission deadline and 6 days before the reviewer's comment. Asking authors to compare with papers released after the submission deadline is considered poor reviewing practice.
> > >
> > > As described in the paper (L123-125), broadly speaking, our work is distinguished from prior literature by a combination of the emphasis on visual perception, lack of access to state, systematic generalization, and the experimental setup of visually-complex and ecologically-realistic home-scale environments.
> > >
> > > More specifically, unlike OpenAI Gym [4], DeepMind control suite [3], Meta-world [1], and RL bench [2], HAB focuses on house scale tasks, with longer tasks (several hundred steps vs 4k steps), a large number of objects, and visual realism [1,2,3,4]. However, HAB does not provide the large diversity of tasks from [1,2] making it less suited for multi-task or few-shot learning.
> > >
> > > The TDW Challenge [5] is more similar since it also involves house-scale rearrangement tasks. As the reviewer mentioned, the TDW challenge abstracts away gross motor control with discrete motion primitives, while HAB uses continuous actions for the base and arm. The motion primitives in TDW use an inverse-kinematics solver to directly reach the goal. This is a simpler setting than our skills which use motion planning or are learned with RL and must succeed in clutter, tight areas, or contacts.
> > >
> > > ManipulaTHOR [6] does not include receptacles in their task. While the AI2-Thor simulator has the ability to open or close receptacles, these state changes are scripted and the robot cannot manually manipulate the receptacle to change its state (for example grabbing the handle to open a door), an important part of realistic interaction provided in HAB. TDW Transport, ManipulaTHOR, and the related RoomRearrangement [7] all use an abstracted high-level action space with actions like “pick object” or “place object”. In HAB agents must learn in a continuous action space for the arm and base. HAB tasks are also longer than prior work at 4k steps compared to ManipulaTHOR (200), RoomRearrangement (250), iGibson [8] (300), or TDW Transport (1k). Sapien [9] interactive benchmarks are limited to opening and closing doors and drawers; they do not involve home-scale assets or tasks
> > >
> > > VirtualHome [10] does not involve interactive tasks and instead executes symbolic programs to script household tasks. DiSECt [11] is a simulator for cutting simulations only (no house-scale tasks) and MegaVerse [12] for fast non-photorealistic simulation in voxelized block-world environments, both different goals from the mobile-manipulation house tasks in HAB.
> > >
> > > **Re3: Bullet Citation and Supported Robots**
> > >
> > > Our paper repeatedly cites Bullet on L112 and L118, and it is credited by name in Table 1 and L238 as the physics engine used in Habitat 2.0. But we are happy to add this reference earlier in the paper in the quoted section.
> > >
> > > Only the Fetch robot is supported in the initial public release, however, we have already run internal experiments with the AlienGo and Franka arm and plan to release these integrations soon. We also provide an extensible framework to add further robot platforms.
> > >
> > > **Re4: Comparison to Bullet’s Island System**
> > >
> > > We will update the paper to describe Bullet's built-in island system and better explain how our Localized Physics contributions complement or build on top of it.
> > >
> > > **Re10: Questions about Time Steps**
> > >
> > > We did not observe simulation artifacts from simulating at 120Hz and based this decision off of other similar simulators such as iGibson which also use the same time steps.  Yes, your clarification is correct, new joint position goals are set at 30Hz and the controller computes joint torques for this same goal and new joint positions at 120Hz. We will update the description in the revised version.
> > >
> > > **Re12: Egomotion Sensing**
> > >
> > > Yes, we use noiseless egomotion sensing. As described in the paper (L268), this is a common assumption in prior navigation works and similar simulators [16,17,18]. Estimating egomotion from RGBD sensors is a challenging research problem on its own [19]. Furthermore, recent techniques for navigation without egomotion sensing show promise as demonstrated in the Habitat 2021 PointNav Challenge [20].
> > >
> > > **Re13: Maximum Instantaneous Force Limit**
> > >
> > > We agree it is useful to add a per time step force limit based on the manufacturers’ suggestions and will do so. Penalizing these instantaneous forces is possible in Bullet.
> > >
> > > Thank you again for your follow-up, please let us know if any additional issues or questions remain.
> > >
> > > **References**
> > >
> > > [1] Yu, Tianhe, et al. "Meta-world: A benchmark and evaluation for multi-task and meta reinforcement learning." Conference on Robot Learning. PMLR, 2020.
> > >
> > > [2] James, Stephen, et al. "Rlbench: The robot learning benchmark & learning environment." IEEE Robotics and Automation Letters 5.2 (2020): 3019-3026.
> > >
> > > [3] Tassa, Yuval, et al. "Deepmind control suite." arXiv preprint arXiv:1801.00690 (2018).
> > >
> > > [4] Brockman, Greg, et al. "Openai gym." arXiv preprint arXiv:1606.01540 (2016).
> > >
> > > [5] Gan, Chuang, et al. "The ThreeDWorld Transport Challenge: A Visually Guided Task-and-Motion Planning Benchmark for Physically Realistic Embodied AI." arXiv preprint arXiv:2103.14025 (2021).
> > >
> > > [6] Ehsani, Kiana, et al. "ManipulaTHOR: A Framework for Visual Object Manipulation." Proceedings of the IEEE/CVF Conference on Computer Vision and Pattern Recognition. 2021.
> > >
> > > [7] Weihs, Luca, et al. "Visual Room Rearrangement." Proceedings of the IEEE/CVF Conference on Computer Vision and Pattern Recognition. 2021.
> > >
> > > [8] Shen, Bokui, et al. "iGibson, a Simulation Environment for Interactive Tasks in Large Realistic Scenes." arXiv preprint arXiv:2012.02924 (2020).
> > >
> > > [9] Xiang, Fanbo, et al. "Sapien: A simulated part-based interactive environment." Proceedings of the IEEE/CVF Conference on Computer Vision and Pattern Recognition. 2020.
> > >
> > > [10] Puig, Xavier, et al. "Virtualhome: Simulating household activities via programs." Proceedings of the IEEE Conference on Computer Vision and Pattern Recognition. 2018.
> > >
> > > [11] Heiden, Eric, et al. "DiSECt: A Differentiable Simulation Engine for Autonomous Robotic Cutting." arXiv preprint arXiv:2105.12244 (2021).
> > >
> > > [12] Petrenko, Aleksei, et al. "Megaverse: Simulating Embodied Agents at One Million Experiences per Second." International Conference on Machine Learning. PMLR, 2021.
> > >
> > > [13] Avetisyan, Armen, et al. "Scan2cad: Learning cad model alignment in rgb-d scans." Proceedings of the IEEE/CVF Conference on Computer Vision and Pattern Recognition. 2019.
> > >
> > > [14] Li et al. "OpenRooms: An Open Framework for Photorealistic Indoor Scene Datasets." CVPR 2021.
> > >
> > > [15] Coumans, Erwin. "Bullet physics library." Open source: bulletphysics. org 15.49 (2013): 5.
> > >
> > > [16] Savva, et al. "Habitat: A platform for embodied ai research." ICCV 2019.
> > >
> > > [17] Chaplot et al. “Learning To Explore Using Active Neural SLAM” ICLR 2020.
> > >
> > > [18] Chaplot et al. “Object Goal Navigation using Goal-Oriented Semantic Exploration.” NeurIPS 2020.
> > >
> > > [19] Datta et al. "Integrating Egocentric Localization for More Realistic Point-Goal Navigation Agents." arXiv:2009.03231 (2020).
> > >
> > > [20] Kadian, et al. "Sim2Real predictivity: Does evaluation in simulation predict real-world performance?." IEEE Robotics and Automation Letters 5.4 (2020): 6670-6677.
> > >
> > > [21] Makoviychuk, Viktor, et al. "Isaac Gym: High Performance GPU-Based Physics Simulation For Robot Learning." arXiv preprint arXiv:2108.10470 (2021).

---

> > > > ### Comment · Reviewer_Xe9G · 2021-09-03
> > > > **Response to authors**
> > > >
> > > > Thank you for your responses.
> > > >
> > > > **Re “Only optimizing physics as the reviewer states...”:**
> > > >
> > > > Reading carefully one could see that I’m not stating that H2 is “only optimizing physics”, but that H2 “accelerates simulation” (that includes rendering), and that is “built around Bullet”. I consider my description correct, the authors’ statement is a misinterpretation.
> > > >
> > > > **Re1,2: ReplicaCAD vs Prior Datasets**
> > > >
> > > > Thank you, this analysis is informative. However, it does not compare in some relevant dimensions like the number of different home layouts, rooms, types of rooms, classes of objects, object models, how many are articulated... very relevant when someone wants to develop a general solution with a simulator. That would make a really interesting comparison and even more valuable information to include in the paper and understand better the value of RepliCAD.
> > > >
> > > > I’m a bit confused about some of the statements. “(VirtualHome [10], IsaacGym [21]) do NOT publicly release their assets “. Does this mean they are not available at all, or that they are not copyright-free? I see the assets from VirtualHome in their repository and I remember trying the code (sorry, I didn’t have time to try it today). They even mention the licenseType as “free” (e.g. https://github.com/xavierpuigf/virtualhome_unity/blob/master/Assets/3DModels/_Appliances/Microwave/Microwave_1.meta) but I assume the authors explored and tested this further to support their reply.
> > > >
> > > > In any case, if this is just about copyright protected vs. copyright-free and not about not being available at all, from a reader/potential user perspective, the copyright issue may not be that important because, often, even copyrighted material can be used for research. If the final scientific product is, for example, a robot policy, I don’t think there are any issues with the copyright of the models, especially for researchers without commercial interests. Furthermore, saying “there is nothing to compare to” just because the assets are not 100% copyright-free is like a paper would present a dataset of 3D shapes and not compare to ShapeNet (if I remember correctly, their copyright is also unclear/not CC): rather incomplete. In a good scientific comparison, mentioning the copyright would be great, but going one step further and comparing the (copyright-free) dataset from H2 vs. the (copyrighted) datasets from others in the most relevant dimensions (number of homes/rooms/objects/articulated objects/soft objects …) would make the analysis complete. This would support and help understand the contribution of RepliCAD and help potential users decide (e.g., do I want to use H2 with “n1” copyright-free models or an alternative with “n2” copyrighted models?).
> > > >
> > > > Just to clarify so that my comments are not again misinterpreted or extracted out of context: I find it very valuable to provide copyright-free (CC) models to the community, I just think an honest comparison could go deeper, even with copyrighted competitors, to give all the information to the readers/potential users. There is already some valuable information in the reply, although no comparison in the number of homes, rooms, objects, classes of objects, …
> > > >
> > > > I’m still not sure how many different homes/rooms layouts are in H2.0, I think it is only one home with multiple furniture arrangements, but I would re-read the paper to try to figure it out for sure.
> > > >
> > > > **Re DiSECt, MegaVerse, and IsaacGym appeared at/after submission. “Asking authors to compare with papers released after the submission deadline is considered poor reviewing practice.”**
> > > >
> > > > Please, read carefully. My sentence in context reads “I agree that it is hard to make very different contributions in this field (although some great recent ones do, e.g., DiSECt [12] or MegaVerse [13]),”. I’m not sure how can someone infer from this that I’m asking to compare to those? I just cited them in the context of my reply as recent examples of simulators that, I consider, make contributions that are clearly distinct from other simulators and, therefore, have an easier time explaining their novelty. To say that I’m “asking the authors to compare” to those is a big stretch that may be seen as a deceitful way to attack the most critical reviewer and devalue his opinion. This is considered poor scientific practice. To be clear: no, it is not necessary to compare to DiSECt or MegaVerse, the differences are obvious and I would not consider them the closest alternatives.
> > > >
> > > > The case of IsaacGym is different. True, the paper I cited appeared recently. However, the software could be tested since December 2020 (https://developer.nvidia.com/blog/introducing-isaac-gym-rl-for-robotics/). Yes, it was not a paper before, but an honest and comprehensive comparison could decide to include it, in my humble opinion. Not a must, most of my recommendations are not, they are just recommendations that, I believe, readers and potential users would consider beneficial to understand better the value of H2.
> > > >
> > > > **Re HAB vs. Prior Tasks**
> > > >
> > > > This is very informative and is the information that is missing in the paper. It would be great to add it to the main paper or the appendix. I can understand HAB better and put it in context after reading this comparison to related work, which is the goal of a good related work section, and I think other readers may find it also very helpful.
> > > >
> > > > **Re “Supported robots”**
> > > >
> > > > If this is more of a “future proof” comment with the expectation that they will be ready soon, I think it is good to have it so that H2 is correctly described at publication. I was afraid it was just random name-dropping, which I understand now is not the case.
> > > >
> > > > With respect to the other replies, they all look good to me, I believe they will improve the manuscript for publication. I thank again the authors for the replies.

---

> > > > > ### Author Response · Authors · 2021-09-07
> > > > > **Response to Reviewer Xe9G**
> > > > >
> > > > > We thank the reviewer for the follow-up response.
> > > > >
> > > > > We first clarify comparisons to relevant datasets in this response. ReplicaCAD is one house with 111 layouts (6 from real scans, 105 artist created), The TDW Transporter Challenge [1] has 15 houses, ManipulaTHOR [2] has 30 rooms, RoomR from Visual Room Rearrangement [3] has 120 rooms, iGibson [4] has 15 houses, and VirtualHome [5] has 6 houses. While house sizes are not reported for these datasets, ReplicaCAD includes kitchens and living rooms, ManipulaTHOR includes kitchens, and RoomR, iGibson, VirtualHome, and TDW include kitchens, living rooms, bedrooms, and bathrooms. ManipulaTHOR and RoomR have single room, not house scale, scenes. In terms of object assets, ReplicaCAD has 92 objects, TDW Transporter challenge has 50 objects, ManipulaTHOR has 150 objects, RoomR has 118 objects, iGibson has 57 objects, Virtual Home has 357 objects, and Sapien PartNet dataset has 2,346 objects.
> > > > >
> > > > > To clarify our comment about VirtualHomes, while some of the VirtualHome assets are at the page linked by the reviewer, based on the released code on GitHub, many of the assets still need to be [purchased from the Unity Store](https://github.com/xavierpuigf/virtualhome_unity/blob/master/doc/third_party.md), and  the assets sourced from PartNet are not released. Thus the entirety of this dataset is not available for comparison. Also, to clarify our comment about IsaacGym [7], IsaacGym is a general physics engine and therefore does not include a dataset of home assets for comparison.
> > > > >
> > > > > We are glad the reviewer acknowledges the importance of our dataset’s license.
> > > > > We further want to stress the importance of proper licensing in creating ML datasets. The history of facial recognition is an example that the practice of scrapping data to train models [is no longer considered acceptable](https://www.nbcnews.com/tech/internet/facial-recognition-s-dirty-little-secret-millions-online-photos-scraped-n981921) without consent from users, even with Creative Commons licensing. In both commercial and non-commercial research, training and re-distributing AI models on data with questionable or unclear licenses should be avoided.
> > > > >
> > > > > Please let us know if any more questions or concerns remain and we are happy to respond.
> > > > >
> > > > > [1] Gan, Chuang, et al. "The ThreeDWorld Transport Challenge: A Visually Guided Task-and-Motion Planning Benchmark for Physically Realistic Embodied AI." arXiv preprint arXiv:2103.14025 (2021).
> > > > >
> > > > > [2] Ehsani, Kiana, et al. "ManipulaTHOR: A Framework for Visual Object Manipulation." Proceedings of the IEEE/CVF Conference on Computer Vision and Pattern Recognition. 2021.
> > > > >
> > > > > [3] Weihs, Luca, et al. "Visual Room Rearrangement." Proceedings of the IEEE/CVF Conference on Computer Vision and Pattern Recognition. 2021.
> > > > >
> > > > > [4] Shen, Bokui, et al. "iGibson, a Simulation Environment for Interactive Tasks in Large Realistic Scenes." arXiv preprint arXiv:2012.02924 (2020).
> > > > >
> > > > > [5] Puig, Xavier, et al. "Virtualhome: Simulating household activities via programs." Proceedings of the IEEE Conference on Computer Vision and Pattern Recognition. 2018.
> > > > >
> > > > > [6] Xiang, Fanbo, et al. "Sapien: A simulated part-based interactive environment." Proceedings of the IEEE/CVF Conference on Computer Vision and Pattern Recognition. 2020.
> > > > >
> > > > > [7] Makoviychuk, Viktor, et al. "Isaac Gym: High Performance GPU-Based Physics Simulation For Robot Learning." arXiv preprint arXiv:2108.10470 (2021).

---

> > > > > > ### Comment · Reviewer_Xe9G · 2021-09-09
> > > > > > **Response to authors**
> > > > > >
> > > > > > Thanks for the replies!
> > > > > >
> > > > > > Yes! This comparison is very informative! Adding that info to the table or somewhere in the paper would be great. Even better, modifying the related work to add this info, since RepliCAD is one of the main contributions. The same about the comparison of the other contribution, HAB, to other sets of tasks.
> > > > > >
> > > > > > Re "assets sourced from PartNet are not released": PartNet-Mobility dataset can be downloaded [here](https://sapien.ucsd.edu/downloads), at least a substantial part of it.
> > > > > >
> > > > > > I think the example used for the argument about copyright is wrong and points towards a different issue: biases in the datasets. The issue with face recognition, as the authors said, is not copyright (commercial issues), the problem is that it handles human data and there are ethical considerations, e.g. biases. Do social media websites have the copyright of the pictures? Are mugshots copyright-free (apparently [yes](https://www.newmediarights.org/business_models/artist/are_mugshots_public_domain))? Should these pictures be used without control and/or permission, even if by the copyright owners or if they are copyright-free? Probably not. At universities, there is the institutional review board (IRB) to check if data and projects involving humans are potentially dangerous. I’m not sure if it is required for companies too. Face recognition falls definitively in this delicate category. Models of objects and furniture do not (no IRB required).
> > > > > >
> > > > > > What is so bad about going to the given [website from VirtualHome](https://github.com/xavierpuigf/virtualhome_unity/blob/master/doc/third_party.md), pay the 35$ for the models, and use them to train agents? I don’t think there is any copyright issue to develop a policy. The 3D models can be used, just not redistributed, even if they are not creative commons. Is this a “questionable or unclear license”? Maybe that was meant for other datasets.
> > > > > > I think we should not demonize copyrighted datasets. They are just not for free, or not to be redistributed, or not to be used for commercial purposes, depending on the license. But beyond the paying part, I don’t think most researchers are affected.
> > > > > > Of course, creative commons licensing is better, as it allows for more things like redistributing and downloading for free (not commercializing, though)! Thanks for doing it! It is valuable if researchers don't have to pay anything, this helps democratize benchmarking.
> > > > > >
> > > > > > A different issue is what 3D models should be in the dataset and how to avoid biases there. For example, are the furniture and objects in RepliCAD based on products of a specific company, or from a specific part of the world, and thus, discriminating others? Who decides what models go into the dataset? How? These are potential biases that models trained in H2 would inherit. But that is what the ethical committee of NeurIPS should evaluate if you think it is necessary; I don't think so. This is the direct translation of the issues in facial recognition to our case, but in any case, these issues of biases are rather independent of the copyright. Maybe the authors could consider mentioning in the paper the “cultural background” the layout and objects in RepliCAD belong to, i.e. western culture?
> > > > > >
> > > > > > Thanks again for the informative replies. This helps to compare RepliCAD to similar datasets.

---

> > > > > > > ### Author Response · Authors · 2021-09-09
> > > > > > > **Response to Reviewer Xe9G**
> > > > > > >
> > > > > > > We thank the reviewer for further engaging in the discussion. We will update the paper with these details of HAB and ReplicaCAD versus prior work. We discuss ReplicaCAD dataset biases in L398-401.

---

### Official Review · Reviewer_ibKn · 2021-07-23

**Rating:** 10
**Confidence:** 5

**Summary:**

This paper presents a simulator for embodied agents for manipulation tasks in house-scale indoor environments. The paper also presents a realistic dataset of household scenarios, and home assistant benchmark (HAB), a suite of tasks designed to evaluate the performance of classical and learning based methods on rearrangement problems.

**Limitations And Societal Impact:**

I liked the limitations and societal impact section -- I believe it touches upon major societal ramifications of the work.

**Main Review:**

Strengths
---------

**S1** Habitat 2.0 (H2.0) represents a significant advance on multiple fronts for the embodied AI community. By curating the ReplicaCAD dataset, providing a simulation package that speeds up training by over 100x, and by introducing a large-scale benchmark of tasks that are currently out-of-reach of deep RL techniques, this paper makes comprehensive contributions to the community and paves way for a potentially large volume of exciting future work.

**S2** The engineering and artistic efforts on ReplicaCAD and H2.0 are impressive. The embodied AI community has a dearth of interactive datasets; existing datasets that are rich are either small scale (AI2-THOR has room-scale scenes) or have very few scenes (iGibson has 14-15 scenes in total). Most simulators that are rich in terms of physics rely on underlying engines (e.g., Unity or Nvidia Flex) and end up being slow. H2.0 provides a highly optimized (CPU) physics engine and interleaves CPU-GPU usage for physics and graphics engines, enabling scaling conventional RL to extremely long training experiences / episodes.

**S3** This paper points out interesting limitations with the combinatorial generalization of learned atomic "skills" (Fig. 5). While atomic skills are learned with high success rates, they do not seem to chain well, and errors from early-on skills in the execution pipeline cascade, often resulting in task failure.

**S4** The work is extremely timely, tackles important yet-unsolved problems, and does so comprehensively. This paper was an unconventional, yet extremely well-written; making it an enjoyable read.


Weaknesses
----------

I have the following concerns that I hope to see discussed in the author response phase.

**W1** *Beyond success rate*: Currently, the sole metric that techniques are being compared upon is "success rate" (i.e., the percentage of test scenarios that end up being 'successful'). While success rate indicates a gross performance of a method, it does not capture the efficiency with which the task is carried out in general. Policies that achieve goals quickly may be preferable to ones that are sucessful, but do so at the expense of additional time. I would like to see more discussion around this in the author response. I believe this to be an important factor given the goal of the benchmark is to spur future work in rearrangement-style tasks involving manipulation.


**W2** *SensePlanAct (SPA) baseline*: Are there any insights as to why SPA-Privileged has a lower success rate in two of the settings in Table 3, when compared to MonolithicRL; despite access to perfect sensing and perfect control? The only (speculative) reason that comes to mind is that bidirectional RRD fails to compute a feasible motion plan before the episode terminates.


**W3** *Solvability*: Is there a mechanism to ensure that each task in the benchmark is "solvable"? If so, is there an "oracle" that would produce a solution to each task? (This might be a useful tool to gather demonstrations for techniques that leverage expert demonstrations)


Minor remarks
-------------

The following comments do not impact my overall evaluation of the paper. I believe these to be easily addressed in a minor revision.

- MonolithicRL: Were subsets of input modalities ablated upon, to analyze the imact of each sensing modality on "pick" success rate? (i.e., how would a depth-only policy compare to a RGB-D policy, etc.)

- Grasping - are there scenarios where unintended grasp releases cause unforseen / unsolvable scenarios? For example, a policy may arbitrarily choose to release an object before it reaches a target receptacle, and the object ends up in a location where it is inaccessible to the end effector.

- SensePlanAct (SPA) baseline: The "sensing" phase is claimed to accumate a 3D pointcloud. How is this constructed? Using an off-the-shelf SLAM system? Does this require relative pose for each input image to be known a priori?


Justification for score
-----------------------

Given the perceived impact such work would have on a significant fraction on the ML community, I would strongly argue in favour of accepting this contribution to Neurips. I believe this work to have immense scope for following-up on, particularly given the promises around permissibly-licensed releases of code and data. While I enlist a few concerns with this work, I believe the strengths comfortably overshadow these concerns, and that it might be difficult to pack more into what is already a content-rich 9-page scientific document.

**Time Spent Reviewing:**

8

---

> ### Author Response · Authors · 2021-08-10
> **Response to Reviewer ibKn**
>
> **1. Why does SPA-Privileged have a lower success rate in two of the settings in Table 3 compared to Monolithic RL; despite access to perfect sensing and control?**
>
> RL given sufficient training experience (100M samples) outperforms SPA-Priv since SPA-Priv uses a finite number of samples in the goal sampling stage. When SPA-Priv cannot find a collision-free joint state as a goal for the motion planner within 1k samples, it fails. We include in-depth analysis into the failure modes of SPA-Priv in Appendix H.1 (L1239) and Figure 26.
>
> **2. Beyond success rate with an efficiency metric**
>
> We agree -- to best enable future work on our proposed benchmark an additional efficiency metric is important. In Appendix E.3 (L958) and Figure 13, we already took a step towards this by reporting the success weighted by completion time (SCT) [1] for the Pick task to measure the efficiency of blind versus sighted policies. SCT reports $S * \frac{T}{\max(C,T)}$ where $T$ is the time duration of an oracle and $C$ the time of an agent. For the Pick tasks, we used an upper bound on the oracle time based on the Euclidean distance between the end-effector and goal.
>
> For the full Home Assistant Benchmark tasks, computing this oracle is more complex as it requires finding the optimal path for the robot base and arm (9D configuration space with 2D for base and 7D for arm) over the 4k step-long episode. Handling these long-horizon task and motion planning problems is an open research problem [2]. Furthermore, optimal solutions can involve contact with the environment for grasping from clutter, opening, or closing, which require controllers that can handle continuous contact and constraints, another open research problem [3,4].
>
> **3. Is there a mechanism to ensure each task in the benchmark is “solvable”?**
>
> We do not yet provide an oracle algorithm to solve the Home Assistant Benchmark tasks. While the task-planning+sense-plan-act (TP+SPA) method utilized oracle scene geometry information it only achieved at most a 70% success rate. Achieving a 100% success rate oracle requires handling continuous contact and constrained full-body planning, both open research problems out of scope for this work [3,4].
>
> However, we manually ensure each benchmark task is solvable. The ReplicaCAD annotated supporting surface data was manually designed by an artist to ensure procedurally placed objects are accessible and not blocked by scene geometry. Furthermore, we implemented a teleoperation setup where the robot is controlled by a human with a keyboard and manually ensured each task is solvable for each scene macro variation.
>
> **4. How is the 3D pointcloud constructed in the sensing phase of Sense-Plan-Act?**
>
> The 3D point cloud is constructed by projecting RGBD pixels into 3D points given the camera intrinsics and extrinsics, derived from the robot base egomotion sensor (line 268). The egomotion sensor gives the coordinates of the robot base relative to the episode starting position. Since the head camera is a fixed transformation from the base, we can project depth camera readings into the relative episodic coordinate system.
>
> **Works Cited**
>
> [1] Yokoyama et al. "Success Weighted by Completion Time: A Dynamics-Aware Evaluation Criteria for Embodied Navigation." arXiv:2103.08022 (2021).
>
> [2] Toussaint, Marc A., et al. "Differentiable physics and stable modes for tool-use and manipulation planning." (2018).
>
> [3] Kingston, Zachary, Mark Moll, and Lydia E. Kavraki. "Sampling-based methods for motion planning with constraints." Annual review of control, robotics, and autonomous systems 1 (2018): 159-185.
>
> [4] Meeussen, Wim, et al. "Autonomous door opening and plugging in with a personal robot." 2010 IEEE International Conference on Robotics and Automation. IEEE, 2010.

---

### Official Review · Reviewer_dawT · 2021-07-25

**Rating:** 8
**Confidence:** 5

**Summary:**

This paper presents the development of a simulator of realistic home environments with robotics support, including physics simulation, photorealism and curated assets. The most relevant novel feature is the increase in simulation speed by orders of magnitude, which enables significantly faster training times. The simulator, which is openly available, also unifies a number of advances introduced by other simulators from the embodied AI community, such as converting real scenes into CAD models for enabling interaction, now into a high performance platform with a well designed and scalable software design. Overall, this paper presents a large engineering effort and investment into developing software for a simulator and environment assets that the community can benefit from.

**Limitations And Societal Impact:**

I don't have concerns here.

**Main Review:**

The paper has the following contributions:

-----(1) ReplicaCAD.----- A dataset of simulation environments created by CAD modelling after real spaces from the replica dataset. These assets are the outcome of extensive design and manual curation, including textures, physical parameters, objects classes, etc, enabling realistic object randomization. The specific idea of producing CAD-based environments is not novel, a point that perhaps deserves to be highlighted more in the paper. That doesn't diminish its importance, as the coupling between the simulator H2.0 and these assets enables comprehensive use of the proposed system. The assets seem to be higher quality than in related work. Clearly the outcome of a large investment as noted in the paper.

-----(2)----- H2.0 high performance photorealistic simulator with physics engine and robot support. To me this is the main contribution of the paper in terms of novel development, as a new tool available to the community and that supports significantly higher simulation speeds. Answering novel research questions might benefit from this speed up in training times. This contribution is mainly covered in Section 4. I would have liked to see even more details on this section, as being the central breakthrough.

* Fig. 3 shows the main point on how authors design concurrent physics stepping and rendering. Since this causes the Observation to be delayed one cycle, I found interesting the argument of biological plausibility (line 214), perhaps it would be interesting to even have this delay being an adjustable parameter.

* I didn't find many details regarding the 1GPU vs 8GPU case. Is the 8GPU case just parallel environments or is it one environment distributed among the 8 GPU? Please offer more details.
* H2.0 takes the approach of keeping physics sim in the processor (using Bullet) and rendering on the GPU, and achieves high speed by using a number of performance optimizations. Another approach is distributed physics simulation in GPUs. For completeness, could the paper cover the pros and cons of these two approaches?

* One of the optimizations is not simulating the physics of contact between the floor and the robot's wheels. I understand the paper claims this has been shown to be a good approximation, but I still worry about the potential of transfering to the real robot. Mixing full physics sim with simplified sim can create a regime where it's hard to know what segments are realistic or not, and therefore hard to debug systems when thinking on transferring to real platforms. Of course we can also think about research purely on the simulator, for which kinematics states are enough to make progress on particular research questions. Same comment applies to the grasping simplification (which is also present in other simulators). While it's true that software-wise this is trivially undone, the main question might be not if H2.0 is able to simulate (actually Bullet), but instead how realistic it is and if that segment of manipulation would behave according to reality.



-----(3)----- Benchmark with household tasks. The paper proposes a small set of well defined robot tasks that can be fully simulated in H2.0. The tasks are mainly instances of the rearrangement problem, a particular way of looking at task and motion planning problems. I think these tasks are a good set of basic problems to support interesting research questions that can be investigated within the simulator. It is challenging to claim how solving the tasks in the simulator would transfer to real robot deployment. This is a broader research question that doesn't diminish the value of the proposed work, but it's worth to acknowledge. The paper also uses the simulator to study algorithms and models for mobile manipulation. From my point of view this section doesn't present a novel contribution in itself, as it treats topics long discussed in other robotics fields, but it's a solid way to verify that the proposed simulator H2.0 is in fact practically useful to investigate these questions.

Other comments:

* I'm not sure using the terminology from the robotics paradigm of sense-plan-act (SPA) to be distinct from the reinforcement learning (RL) policy is the best way to highlight how these two overall approaches are different. In SPA, you refer to the pipeline sensor to motion planner to controller, and in some sense a policy also has a sensor input, to forward pass of the policy, to execution (sometimes policy output is still given to a similar low level platform controller). In robotics SPA refers to the overall flow of information and decision making (from sensor, to decision, to action execution). The most frequently used SPA uses a "motion planner", while RL uses the "forward pass of a neural network policy). This is just a thinking point about the terms.

* The paper is very clear and the details and board set of considerations are well thought. The writing makes frequent use of parentheses, which seemed too much but actually made the writing very concrete, grounding the general comments into specific aspects that people in the field will get right away.

* While the presented experiments show the usability of the simulator and assets for this type of problems, I think it would also be important to report more quantitatively what quality to expect from the physics simulation in comparison with the real world objects. The paper reports to have authored the physical parameters of the assets but just from reading the paper I'm not sure what level of realism to expect if I wanted to use H2.0 for robotics research.

* One of the optimizations for speeding up the simulation is to use object sleeping for objects beyond a certain distance from the robot agent. How can the approach be adapted to multi-agent systems? For example, more than one robot in the scene.

* I'm not sure if this paper should be on the main paper submission track on in the benchmark track.

* The video should have been submitted with supplementary materials, not only a link on the paper.


**Time Spent Reviewing:**

5

---

> ### Author Response · Authors · 2021-08-10
> **Response to Reviewer dawT**
>
> **1. Mixing full and simplified physics makes it hard to know what segments are realistic or not, and hard to debug systems when transferring to real platforms**
>
> We generally agree with the risk that the reviewer is highlighting. However, we believe (a) that the benefits of significantly increased simulation speed make the risks worth undertaking, and (b) that these risks can be mitigated via analyses that isolate the different segments. Specifically, we know from prior work in navigation [6,8,11] that policies trained with the navigation mesh abstraction transfer well to reality. Our abstracted grasping is consistent with the point-click grasping APIs provided by modern mobile manipulators such as SPOT by Boston Dynamics (https://dev.bostondynamics.com/).
>
> In general, our operating hypothesis is this -- we can train "high-level" policies (or gross motor control) in simulation and pair them with low-level controllers or fine-motor control (powering wheel motors, grasping an object) provided by real robotic platforms. This means that we do not have to solve the hard problem of accurately modeling low-level physics of the world as an intermediate step to performing sim2real.
>
> This choice is validated in our experiments, where even with abstracted grasping, all methods performed poorly on the full tasks (Figure 5). Furthermore, in the rebuttal response to Xe9G, Pdk8, we remove the abstracted grasping and show results with full grasping.  Even with the full grasping, picking policies learn with end-to-end RL, although to a lower success rate. A more thorough investigation of success in real-world transfer is a goal we leave outside our scope.
>
> **2. Would be interesting to have the observation delay be an adjustable parameter**
>
> We agree and have implemented this suggestion. To concurrently render and step physics, Habitat 2.0 delays observations by 1 cycle, meaning the policy is changed to $\pi(a_t | o_{t-1})$ from $\pi(a_t | o_t)$. We adjust the size of this delay with policies $\pi(a_t | o_{t-i})$ from $i = 0$ (real-time), $i=1$ (our experiments), to $i=2,3,5$. We find that a one and zero-step delay perform near-identically (89.83% versus 89.70%).  Larger delays produce worse performance with the policy not having enough time to react with a 2 step delay succeeding 85.86% of the time, 3 step 73.17%, and 5 step 68.33%. Note that a one-step delay is sufficient to achieve performance speed-ups through concurrent rendering and physics. These experiments were run in the setting from Appendix E.2 with two seeds. We will add this analysis to the supplement. Thank you for the suggestion!
>
> **3. Could the paper cover pros and cons of distributed physics simulation with GPUs**
>
> GPU physics simulation is typically faster than CPU physics simulation when the amount of simulation work exposed to the GPU is large, either in scenarios that are naturally highly-parallelizable such as cloth or fluid physics or, in scenarios like H2.0 with a relatively small number of rigid/articulated objects, by combining the work of many environments together via batched physics simulation [9, 10]. Batched physics and rendering are currently in their infancy. While we see them as the future, they don’t yet offer the same flexibility and feature set as non-batched systems and thus aren’t yet ready for research platforms like H2.0 where flexibility is important.
>
> **4. How can the physics object sleeping optimization be extended to multiple agents?**
>
> The short answer is that no adaptation is needed.  Our object-sleeping implementation doesn't assume a single agent: a sleeping object is awakened when its bounding box overlaps with a non-sleeping object's bounding box. E.g. a (non-sleeping) robot arm moves into proximity with a (previously-sleeping) cup. This logic will work for a scene with multiple robots. However, the simulation speed would decrease because fewer objects would sleep.
>
> **5. Details of 1GPU vs 8GPU benchmark settings**
>
> We provide these details in Appendix B.3 (line 787). In the 8GPU case, 64 processes (a process is the same as a parallel environment) were used with 8 processes assigned to each GPU.
>
> **6. The specific idea of producing CAD-based environments is not novel**
>
> We agree and will highlight other efforts such as Scan2CAD [1] and OpenRooms [2]. Please feel free to suggest anything else.
>
> **Works cited**
>
> [1] Avetisyan, et al. "Scan2cad: Learning cad model alignment in rgb-d scans." CVPR 2019.
>
> [2] Li et al. "OpenRooms: An Open Framework for Photorealistic Indoor Scene Datasets." CVPR 2021.
>
> [3] Savva et al. "Habitat: A platform for embodied ai research." ICCV 2019.
>
> [4] Xia et al. "Gibson env: Real-world perception for embodied agents." CVPR 2018.
>
> [5] Chang et al. "Matterport3d: Learning from rgb-d data in indoor environments." 3DV 2017.
>
> [6] Kadian et al. "Sim2Real predictivity: Does evaluation in simulation predict real-world performance?" RA-L 2020
>
> [7] Chaplot et al. “Learning To Explore Using Active Neural SLAM.” ICLR 2020.
>
> [8] Chaplot et al. “Object Goal Navigation using Goal-Oriented Semantic Exploration.” NeurIPS 2020.
>
> [9] Freeman et al. “Brax - A Differentiable Physics Engine for Large Scale Rigid Body Simulation.” NeurIPS Datasets and Benchmarks Track 2021
>
> [10] IsaacGym, NVIDIA, 2020 https://developer.nvidia.com/isaac-gym
>
> [11] Saurabh Gupta et al. Cognitive Mapping and Planning for Visual Navigation IJCV 2019

---

> > ### Comment · Reviewer_dawT · 2021-09-03
> > **Post-response**
> >
> > Thank you for the response and clarifications. I have read all the points.
> >
> > (1) The point of whether or not we need simulators for robot learning that solve for the full low level details including physics and contact is frankly an open question, at least for the case of transferring to real robots in unstructured environments. I think this abstraction can work for some tasks, not all tasks, and perhaps not for manipulation in general.
> >
> > (2) I find interesting the additional experiments regarding the observational time delay. I think this point is important for successful transfer to real hardware as well.
> >
> > (3)(4)(5) Ok.
> >
> > (6) As mentioned by other reviewers as well, the writing could, and should do a better job at highlighting the source of some ideas. The paper is not *missing* citations, but it's not *using* them in key sentences, in a way the the informed reader notices. For this example of CAD-based environments, you bring up Scan2CAD [1] and OpenRooms [2]. Those concern the cad modelling aspect, but relevant to this paper is the idea of using those for creating virtual environments that resemble realistic human environments for the purposes of robot training.
> >
> > The current surge of many efforts to develop simulation for robotics and embodied AI responds to the underlying believe that better simulations would help us answer novel and challenging research questions. As to what degree simulator A or B will accomplish this goal only time will tell, but for sure the plurality of efforts will add up to make overall progress in the community until there is some convergence to the best tools and platforms, which would make it easier for researchers to share work, benchmark, reproduce and compare.

---

> > > ### Author Response · Authors · 2021-09-07
> > > **Response to Reviewer dawT**
> > >
> > > We thank the reviewer for the response. We will update the paper to also use the citations in key sentences so readers can better understand the relation between our work and prior work.

---

### Author Response · Authors · 2021-08-10
**Response to all Reviewers**

We thank the reviewers for their insightful feedback. We are excited to see a predominantly positive response to our contributions: the ReplicaCAD dataset, the Habitat 2.0 simulator, and the Home Assistant Benchmark (HAB) task suite. Together, we hope these contributions will enable the community to answer new research questions at larger scales of simulator experience in interactive 3D environments.

This sentiment is echoed by three reviewers: “simulator and environment assets that the community can benefit from” (dawT), “high performance platform with a well designed and scalable software design” (dawT), “represents a significant advance on multiple fronts for the embodied AI community” (ibKn), “makes comprehensive contributions to the community and paves way for a potentially large volume of exciting future work” (ibKn), “building up the pipeline and platform to carry out tasks at extreme high frame-rates is very commendable” (Pdk8).

We are thankful for reviewer suggestions and will incorporate them in revisions. We answer specific reviewer questions through separate individual comments.

In this high-level comment, we would like to address head-on the questioning of the value of engineering in science, as illustrated by Xe9G’s summary comment:

“In summary, I see the engineering value of the work, but the scientific value and the intellectual novelty are not high.”

We must call out the strong implicit bias in this statement. It comes from viewing engineering pejoratively and infantilizing its contribution to scientific progress, which is wrong at multiple levels.

Where would the deep learning community be without the engineering tools (Caffe, Theano, Tensorflow, PyTorch), the datasets (ImageNet, COCO, SNLI, VQA, SquaD), and associated benchmarks?

We embrace the label of engineering with pride. Our work is on big science enabled by large-scale engineering.

As dawT says “The Habitat 2.0 simulator is a new tool available to the community and that supports significantly higher simulation speeds”.

The 100x speedups over prior work dramatically change what is possible: an experiment that would previously take 6 months can now be carried out in 2 days. And this is now available to everyone. That is the impact of our work.

As ibKn says: “Given the perceived impact such work would have on a significant fraction on the ML community, I would strongly argue in favour of accepting this contribution to Neurips. I believe this work to have immense scope for following-up on, particularly given the promises around permissibly-licensed releases of code and data.”

The scientific value of our work is that it will enable the next generation of research in mobile manipulation, reinforcement learning, and embodied AI.

---

### Decision · Program_Chairs · 2021-09-27

**Decision:**

Accept (Spotlight)

**Comment:**

The paper presents the second version of a well-known and well used simulator, environment and benchmark for embodied computer vision, in particular navigation tasks.

The paper received 4 expert reviews with a wide spread (minimum rating 4, maximum rating), and was initially on the fence. The main weaknesses raised where a lack of methodology, as the paper mainly describes an engineering contribution.

In the discussion phase, a consensus emerged on the usefulness of this kind of contribution for the community, as the impact of the new simulator can be estimated to be large.

The AC concurs and recommends acceptance.